evolution, microbiology, theoretical biology

kin selection, group selection, multilevel selection, microbial cooperation, open data, open science

**Author for correspondence:**
Jeff Smith
e-mail: matryoshkev@gmail.com

# Evaluating kin and group selection as tools for quantitative analysis of microbial data

Jeff Smith and R. Fredrik Inglis

Department of Biology, University of Missouri–St Louis, St Louis MO 63121, USA

JS, 0000-0001-8445-586X; RFI, 0000-0002-3986-7256

Kin selection and multilevel selection theory are often used to interpret experiments about the evolution of cooperation and social behaviour among microbes. But while these experiments provide rich, detailed fitness data, theory is mostly used as a conceptual heuristic. Here, we evaluate how kin and multilevel selection theory perform as quantitative analysis tools. We reanalyse published microbial datasets and show that the canonical fitness models of both theories are almost always poor fits because they use statistical regressions misspecified for the strong selection and non-additive effects we show are widespread in microbial systems. We identify analytical practices in empirical research that suggest how theory might be improved, and show that analysing both individual and group fitness outcomes helps clarify the biology of selection. A data-driven approach to theory thus shows how kin and multilevel selection both have untapped potential as tools for quantitative understanding of social evolution in all branches of life.

## 1. Introduction

For biologists seeking to understand the evolution of cooperation and social behaviour, mathematical theory has been a valuable guide for empirical research. Kin selection theory, for example, has been enormously influential in part because it shows that social evolution can be understood from just a few important quantities: how a behaviour affects the fitness of an actor, how that behaviour affects the fitness of other individuals, and the genetic relatedness between social partners [1,2]. This theory was developed with a major focus on animal behaviour, so its mathematical form was developed in terms of the quantitative-genetic phenotypes most easily measured in animal systems [3]. Similarly, evolutionarily stable strategy (ESS) models of kin selection were formulated to facilitate predictions and empirical tests using comparative data [4] with substantial success [5–7].

More recently, microbial systems have become popular for addressing social evolution [8]. Aside from these organisms' intrinsic biological interest and importance as pathogens and mutualists, they allow for empirical methods that would be impractical or impossible in non-microbial systems. Fitness, for example, is notoriously difficult to measure in multicellular organisms [9]. In microbial systems, though, it is common practice to directly measure the fitness effect of a defined change in genotype or environment under controlled laboratory conditions. Evolution can even be observed as it happens and replayed under different conditions [10,11].

Microbial systems also have their challenges. It can be challenging to measure or even identify the phenotypes responsible for some fitness effect. And since meiotic sex is often infrequent or absent, studies of microbial genetics tend to focus on the effects of defined, discrete genotypes instead of quantitative genetic analyses of continuous-valued traits (but see [12]). It can also be difficult to study microbial behaviour in natural environments, making comparative tests of ESS

Proc. R. Soc. B **288**: 20201657

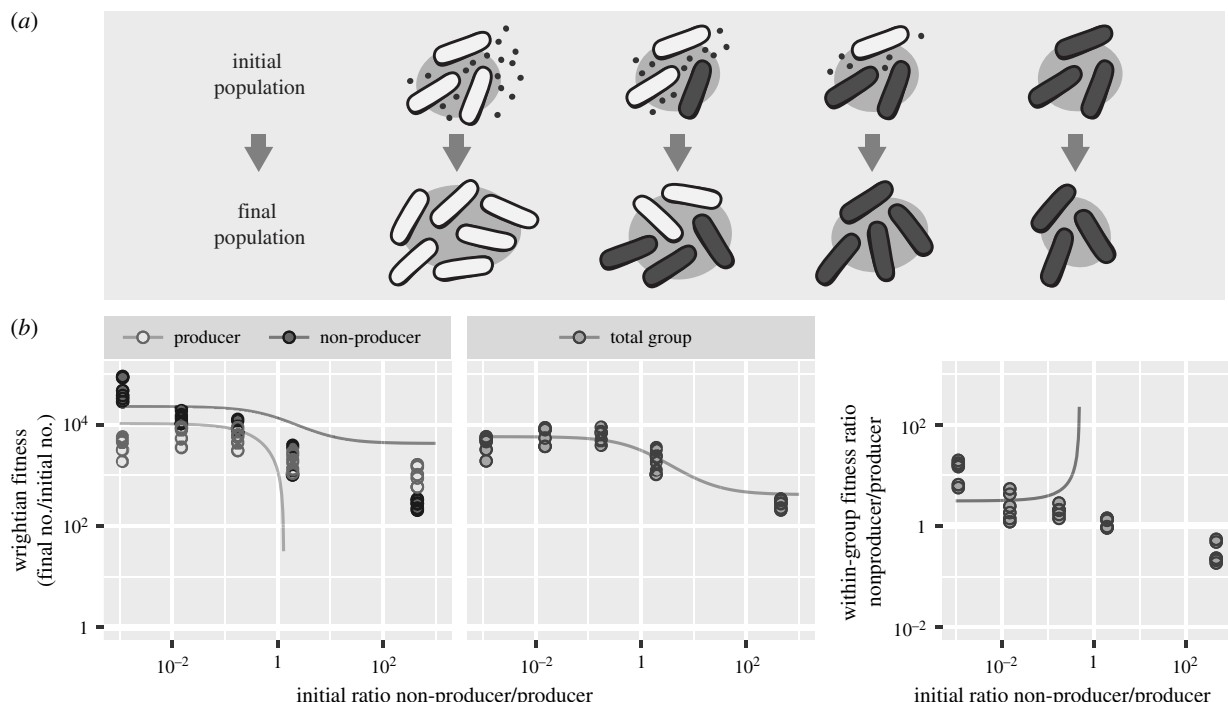

**Figure 1.** (a) Mix experiments investigate the fitness effects of microbial interactions by manipulating local genotype frequency. Illustration shows design of a mix experiment for a hypothetical 'cheating' interaction where a strain benefits from an external good without paying the cost to produce it. (b) The canonical versions of kin and multilevel selection both use statistical regression models poorly specified for the strong selection and non-additive fitness effects in microbial data. Plots show example dataset from *Pseudomonas* bacteria that do or do not produce a secreted iron-chelating siderophore (PAO1 + PAO6609) [27]. Lines show fit of canonical fitness models, whose additive fitness effects mischaracterize data and predict nonsensical results like negative cell number. Non-additive effects include strains responding differently to mixing- and frequency-dependent selection within groups.

models rare. So the mathematical formulations of social evolution theory used in metazoan research are poorly suited to the types of data microbiologists most often collect. Perhaps as a result, microbial researchers often use *ad hoc* methods to analyse and interpret their data, only using theory as a conceptual heuristic or source of qualitative predictions.

Microbes are not intractable to mathematical theory, however. In the epidemiology of infectious disease, for example, there is extensive close interaction between models and data, with models formulated to make predictions in terms of the data empiricists collect [13,14]. Models are tailored to the richness of available data and then used to make quantitative predictions. Deviation of data from these predictions is then used to refine models in an iterative cycle of improvement. One of the most important results to come out of this theory is a summary statistic called $R_0$: the average number of secondary infections from a single infected individual in a wholly susceptible population [13–15]. $R_0$ provides an estimate of outbreak population dynamics and pathogen fitness, allows researchers to compare very different infectious agents, and guides public health responses to diseases like SARS [16,17], Ebola [18] and COVID-19 [19].

What would a mathematical formulation of social evolution look like if it were able to engage with the quantitative details of data from microbial experiments? Perhaps the weak, additive fitness effects common in canonical theory [2,3,9,20,21] are problematic in microbial systems where large populations and rapid reproduction make selection especially effective. Is kin selection theory the best approach, or would it be more promising to use multilevel selection, which examines how selection acts within and between groups of interacting individuals [22–24]? The proper formulation of

social evolution theory has been contentious [25,26], but it has been debated mostly as abstract mathematics or as verbal interpretations of specific models and results.

Here, we evaluate kin and multilevel selection as tools for quantitative analysis of microbial data. We focus on a common experimental design we call a 'mix experiment' (figure 1a). Mix experiments investigate the fitness effects of microbial interactions by manipulating local genotype frequency. One can, for example, compare the fitness of genotypes when they are separate to their fitness when they are allowed to interact. Fitness is typically measured by comparing the initial and final states of an experimental population. In practice, mix experiments examine how selection acts in a local subpopulation during one part of a microbe's life cycle—within a single infected host, for example. Their results can then be used to estimate how selection in a larger metapopulation would depend on genetic social structure at this scale (the distribution of initial genotype frequencies among subpopulations). Mix experiments do not typically measure microbial behaviour or other social phenotypes directly.

We conduct kin and multilevel selection analyses of datasets from published mix experiments and use quantitative measures of statistical performance to assess what these approaches do well, where they run into problems, and how often these problems occur. For guidance on how theory might better handle the challenges of microbial data, we also identify analytical practices in empirical research that are robust across different microbial systems, provide insight into the causes of selection, and allow quantitative comparison of social selection across systems. A quantitative, data-driven approach can be a productive way forward to identify how theory can best aid our understanding of social evolution in all branches of life.

## 2. Methods

### (a) Datasets

We surveyed published literature for mix experiments with different genotypes of the same microbial species, focusing on studies of social evolution. We included experiments published in the last 20 years that measured the asexual survival and reproduction of strains as a function of their initial frequency, holding constant the total number of individuals. We aimed for breadth of species and social trait, prioritizing studies with several mixing frequencies (initial genotype frequencies) over those with a single 50 : 50 mix. We identified 39 studies with experiments matching our criteria. We sought raw data from journal websites, public data archives and directly from study authors. So that our results would not be overly dominated by a few studies with many experiments, we included a maximum 12 datasets per study, prioritizing those with most replication and covering the observed range of interactions. In all, we obtained 80 datasets from 20 different studies for further analysis (electronic supplementary material, table S1).

### (b) Applying theory

Taking mix experiments as assays for the fitness effects of interactions within a single group of cells or virions, we let a strain's Wrightian fitness $w$ be its final absolute abundance divided by its initial abundance (electronic supplementary material, equation (S3)) and let its Malthusian fitness be $m = \ln w$. $w$ and $m$ measure the net performance of a strain over the entire course an experiment, potentially including multiple rounds of reproduction. Electronic supplementary material, table S2 summarizes our notation.

Kin and multilevel selection are different frameworks for understanding natural selection when individuals interact. For the same biological situation, these theories use different terms to describe which individuals interact and how those interactions affect fitness outcomes. Because social phenotypes are not typically not measured in mix experiments, we use theory formulated in terms of genotypes and fitness. We call these formulations 'canonical' because they are among the most general descriptions of social selection and are common touchpoints in theoretical discussion.

A common form of kin selection theory is neighbour-modulated fitness, which describes fitness as a function of one's own genotype and that of one's social partners [2,28]. If strain genotype is $g$, with $g = 1$ for the focal strain of interest and $g = 0$ for its partner, and $G$ is mean group genotype (the initial frequency of the focal strain), then kin selection canonically describes fitness using a multiple least-squares regression of the form $w \sim g + G$ (electronic supplementary material, figure S1 and equation (S6)). The slope of $w$ with respect to $g$ is the direct effect of individual genotype (the private cost of cooperation, for example), and the slope of $w$ with respect to $G$ is the indirect effect of neighbour genotype (e.g. the shared benefit of cooperation).

A common form of multilevel selection theory uses the Price equation to describe fitness within and between groups [22,23]. If total group fitness is $W$ and the within-group fitness difference between strains is $\Delta w$, then multilevel selection canonically describes fitness using least-squares regressions of the form $W \sim G$ and $\Delta w \sim 1$ (electronic supplementary material, figure S1 and equation (S9)). The slope of $W$ with respect to $G$ is the among-group selection gradient, and mean $\Delta w$ is the within-group gradient.

### (c) Calculations and statistics

We analysed all data using R [29]. We assigned wild-type and ancestral strains to reference genotype and mutants and laboratory-evolved derivatives to focal genotype. We fitted statistical models by least-squares regression (`lm()` command), excluding zeroes for comparison involving log-transformed quantities.

We evaluated model performance using the Akaike information criterion (AIC) [30,31] calculated via `AIC()`. AIC measures goodness of fit discounted by the number of model parameters. $\text{AIC} = 2k - 2 \ln \hat{L}$, where $k$ is the number of parameters and $\hat{L}$ is maximum likelihood. AIC comparisons are meaningful even when models are not nested and have different predictors. A common rule of thumb is models perform substantially differently if $|\Delta \text{AIC}| > 2$, equivalent to adding a parameter without improving goodness of fit. The best-performing model is the one with the least AIC.

We sometimes compared the best-performing models (least AIC) from sets of candidates (see electronic supplementary material, table S3 for model details). To compare models fit to linear or log-transformed fitness outcomes, we calculated the AIC of logarithmic models, as measured on the linear scale, as `AIC()` plus the Jacobian of the log transform: $2 \sum \ln y_j$, where the $y_j$ are the fitted values [32]. To compare models fit to different fitness outcomes, we calculated predicted values for the other outcome set (electronic supplementary material, equations (S11) and (S12)) then fit a Gaussian error parameter via maximum likelihood (`mle()` command) so that models would have the same error structure. We counted $M$ and $\Delta m$ as part of the same model so that neighbour-modulated and multilevel fitness models each used four parameters and $\Delta \text{AIC}$ measured relative goodness of fit.

To compare fitness effect sizes, we fit 'log-linear' models of the form $m \sim g + G + gG$, $M \sim G$ and $\Delta m \sim G$, where slopes are comparable to intercepts because they measure the total effect of group genotype from zero to one. We calculated unstandardized effect sizes as the regression coefficients $\beta$ in fitted models and scaled effect sizes as $\beta' = \beta / \sum |\beta|$, where the sum includes all four coefficients.

## 3. Results

To evaluate social evolution mathematics as tools for analysing microbial data, we performed kin and multilevel selection analyses of published datasets that share a common 'mix experiment' design (figure 1a). We found that the canonical fitness models used by both theories were almost always poor choices for extracting quantitative summary statistics because they used statistical regressions misspecified for the strong selection and non-additive effects we found were widespread in microbial systems.

### (a) Strong selection is widespread in microbial data

Kin and multilevel selection are canonically formulated in terms of Wrightian fitness—the number of individuals in the descendant population per individual in the ancestral population (electronic supplementary material, equation (S3)). Measured over the entire course of a microbial mix experiment, this value can span several orders of magnitude (e.g. figure 1b). As a result, the additive fitness models used by both theories mischaracterize data in regions of low fitness and predict nonsensical results like negative cell number. The linear scale over which these theories describe social selection is also a poor choice for visualizing data because many values cluster near zero (electronic supplementary material, figure S2).

To assess how common strong fitness effects are in microbial systems, we measured the range of fitness values in published datasets (electronic supplementary material, figure S3). Mixing frequency typically affected strain and

Proc. R. Soc. B **288**: 20201657

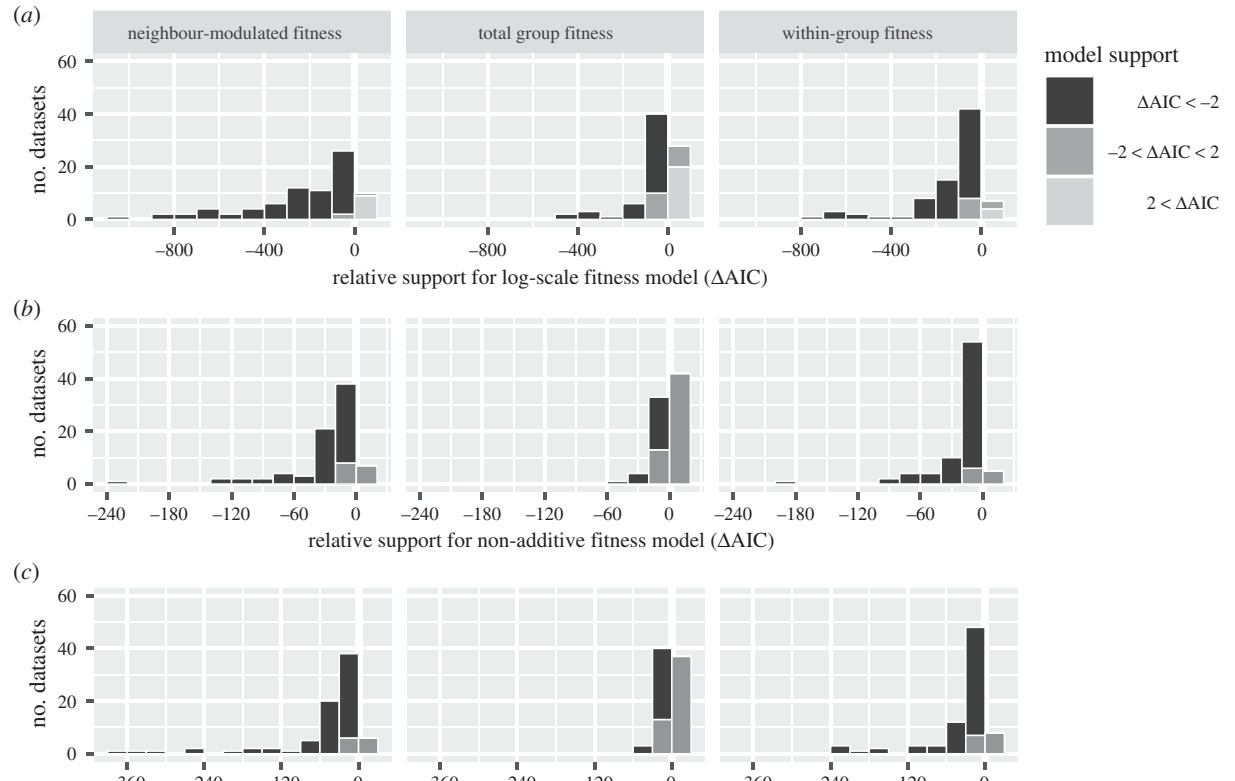

**Figure 2.** Fitness models usually perform better (least AIC) when they allow strong selection and non-additive effects. (*a*) Analysing data over logarithmic instead of linear scales (e.g. Malthusian instead of Wrightian fitness) improves model performance. (*b*) Over linear fitness scales, allowing non-additive effects improves model performance. (*c*) Even over logarithmic fitness scales, allowing non-multiplicative effects improves model performance. See electronic supplementary material, table S3 for model details.

total group fitness by orders of magnitude. Within groups, the fitness of one strain was often orders of magnitude greater than the other. In most datasets, strong selection was the norm. We even found strong effects in experiments measuring the survival of starving *Myxococcus* bacteria to become spores in fruiting bodies, with little to no reproduction.

Many studies dealt with strong selection by analysing experimental data over logarithmic scales (electronic supplementary material, table S4). To evaluate this approach, we compared models fit to linear or log-transformed fitness outcomes with the same predictors. Log-scale models almost always outperformed linear models (figure 2*a*).

### (b) Non-additive fitness effects are widespread in microbial data

Kin and multilevel selection are canonically formulated with additive fitness effects, corresponding to statistical models with linear, independent effects on Wrightian fitness (electronic supplementary material, figure S1). In the datasets we analysed, fitness effects were often strongly non-additive (e.g. figure 1; electronic supplementary material, S2). Fitness responded nonlinearly to mixing frequency, for example, there were interaction effects between self and neighbour genotype, and there was frequency-dependent selection within groups (within-group fitness was a function of initial genotype frequency). As a result, kin and multilevel fitness models were frequently poor fits.

To quantify the extent of non-additivity, we compared additive models to those with non-additive terms. Non-additive models almost always performed better for

neighbour-modulated and within-group fitness (figure 2*b*). For total group fitness they performed at least as well. In most datasets, non-additivity was the norm.

Strong selection can create fitness effects that are multiplicative rather than additive, with independent linear effects over logarithmic scales. To assess how much this contributed to non-additivity, we compared multiplicative models to those that could include interaction and nonlinear terms. Nonmultiplicative models almost always performed better (figure 2*c*).

Model performance was particularly improved by allowing frequency-dependent selection within groups and interactions between individual and neighbour genotype (electronic supplementary material, figure S4). These additional terms were not just minor complications—they were typically a substantial proportion of all fitness effects and could be the strongest social effect (figure 3; electronic supplementary material, S5 and S6). Initial frequency often determined which strain was more fit (e.g. figure 1*b*).

### (c) Strain and group fitness outcomes are both useful

Kin and multilevel selection focus on different fitness outcomes. For microbial mix experiments, kin selection focuses on strain fitness, while multilevel selection focuses on total group fitness and the relative fitness of strains within groups (electronic supplementary material, figure S1). What are the consequences of analysing a particular set of outcomes?

In the datasets we examined, we found cases where one set of fitness outcomes revealed some biological result the other did not. Analysing data from multiple perspectives

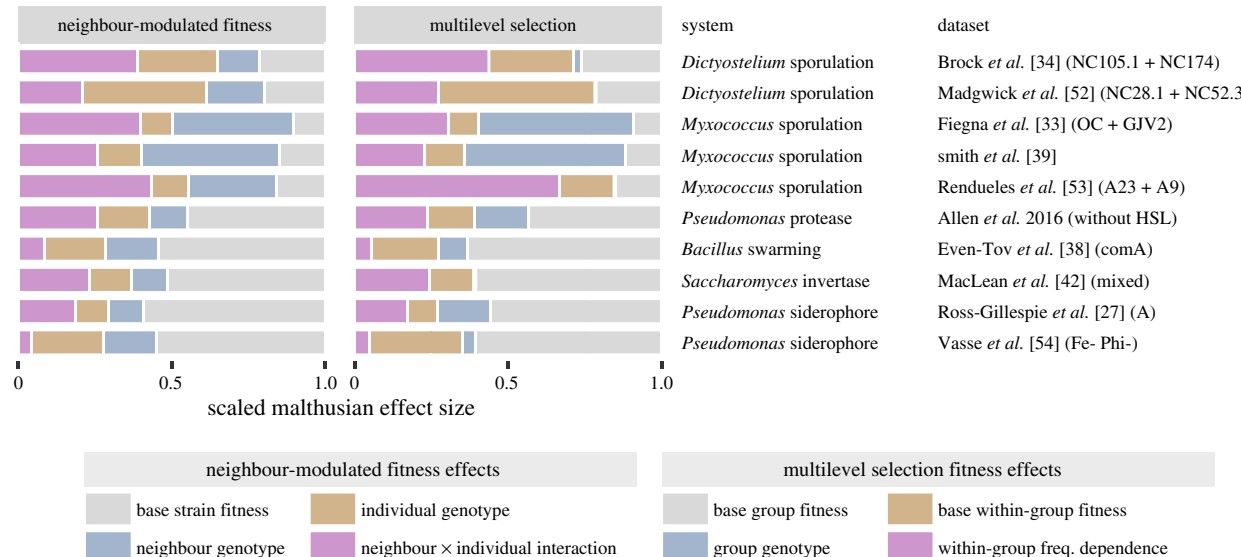

**Figure 3.** Nonstandard fitness effects are often strong (shown in pink, neighbour x individual interaction and within-group freq. dependence). Also, social interactions have more impact in some systems than in others. In many *Dictyostelium* and *Myxococcus* datasets, for example, most fitness effects are social. In other datasets, the largest effect is base fitness. Data show Malthusian effect sizes for select datasets, scaled as a fraction of all observed effects. Colours indicate analogous effects. (Online version in colour.)

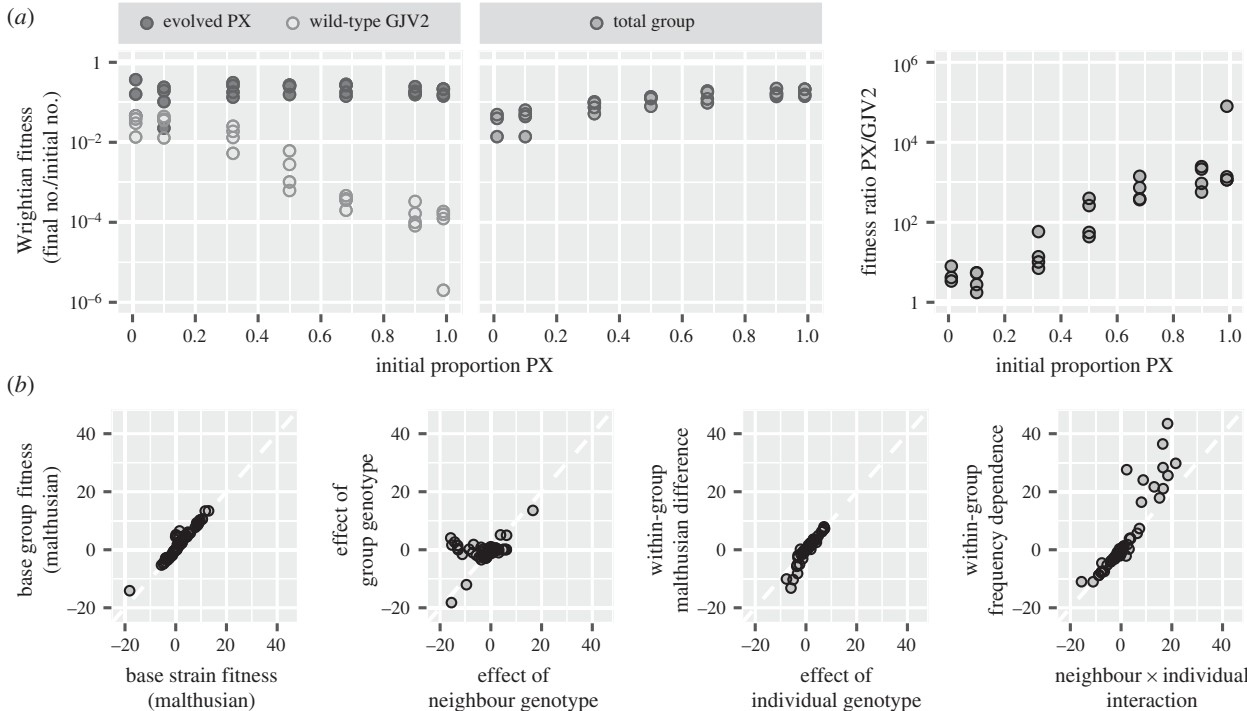

**Figure 4.** Analysing both strain and group-focused fitness outcomes can reveal results that would otherwise go unseen. (*a*) Example from *Myxococcus* bacteria. Mixing reduces the fitness of wild-type strain GJV2 but has no effect on laboratory-evolved strain PX. This one-way effect is not apparent in the original study [33] because it only reports multilevel fitness outcomes. (*b*) Across datasets, kin and multilevel selection analyses mostly tell similar stories, but mixing frequently affects strain fitness much more than total group fitness. Data compare analogous Malthusian effect sizes in models fit to the same dataset.

also revealed similarities between systems that would not have been otherwise visible. During multicellular sporulation of *Myxococcus* bacteria, for example, evolved strain PX inhibits sporulation of its ancestral wild-type but is itself unaffected by mixing (figure 4*a*). Researchers were aware of this one-way mixing effect (G. J. Velicer 2019, personal communication), but because they only reported multilevel fitness outcomes it is not apparent in the original study [33], nor is its similarity to later findings in *Dictyostelium* amoebae (electronic supplementary material, figure S16) [34].

To quantify the extent to which effects seen in one set of fitness outcomes were apparent in the other set, we fitted neighbour-modulated and multilevel fitness models to each dataset and then compared analogous effect sizes. Strain and multilevel fitness mostly told similar stories, with the notable exception that mixing frequently affected strain fitness much more than total group fitness (figures 4*b*; electronic supplementary material, S6). In many datasets, there was strong selection within groups but little change in total group fitness, a situation known as soft selection [24,35]. As a result, multilevel selection could consistently describe data using fewer effective

parameters (electronic supplementary material, figure S7). Some datasets were easier to analyse with strain fitness and others with multilevel fitness, but there was no strong trend (electronic supplementary material, figure S8).

Strain and multilevel fitness outcomes were both effective for quantitatively comparing social selection in different datasets. The impact of social interaction varied considerably among systems (figure 3; electronic supplementary material, figure S5). In datasets concerning multicellular sporulation of *Myxococcus* or *Dictyostelium*, for example, most fitness effects were social. The effects of shared external goods like siderophores or secreted enzymes, on the other hand, were often minor compared to base strain or group fitness.

Together, these results suggest it is empirically useful to analyse both strain and multilevel fitness, then use whatever is most convenient or informative. The studies we analysed most often reported multilevel fitness (electronic supplementary material, table S4 and figure S9), even those verbally framed in terms of kin selection [27,36–38]. Multilevel selection thus appears to hold substantial intuitive appeal, and there is scope for more extensive use of strain fitness data.

## 4. Discussion

### (a) Disconnect between theory and data

To evaluate kin and multilevel selection theory as tools for analysing data about microbial cooperation and social behaviour, we reanalysed the results of published mix experiments using both approaches. We found that while the different fitness outcomes emphasized by each theory were both useful, the canonical fitness models in both theories were poor choices for quantitative analysis and comparison that were almost always a poor fit for the strong selection and non-additive effects that we found were widespread in microbial data.

The fitness terms in kin and multilevel selection are statistical regressions made over the individuals and groups in the population they are meant to describe. This approach is general and exact even when fitness effects are not weak or additive [2]. We are not claiming that kin or multilevel selection theory are conceptually incorrect, nor that they make incorrect assumptions. But when the regression models do not match the relationships in the data, the regression coefficients will not be true for other populations with a different genotype frequency or a different distribution of genotypes among groups, even if eveything else is the same [21,39,40]. So the fitness terms in these misspecified regressions are not very useful summary statistics for quantifying and comparing social selection in microbial systems. It is perhaps unsurprising then that empirical microbiologists use other methods to analyse and interpret their data.

Non-additivity is an active area of research in social evolution theory [2,3,21]. Non-additivity may not be a problem for the weak selection and qualitative predictions common in metazoan research [2,9,20], but it has caused problems for attempts to measure the parameters of Hamilton's rule in microbial systems [39,41]. Nonlinear responses are common in microbiological research [27,42,43], but *ad hoc* data analyses have so far made it difficult to assess what implications this has for theory. By analysing microbial datasets using the same quantitative fitness models as theory, we found that the problems posed by non-additivity are widespread, strong, and beset both kin and multilevel selection.

The culprit was the functional form of fitness models, not the fitness outcomes themselves. Analysing strain fitness (like in kin selection theory) or fitness within and between groups (like in multilevel selection theory) are both useful approaches to microbial data. Different fitness outcomes can reveal results that would not be otherwise visible. Which approach is best will depend on the system and question being investigated, and may not be obvious beforehand. So while the proper formulation of social evolution theory is sometimes contentious [25,26], it is practically useful to examine data from multiple perspectives [44]. An analogy might be how physics embraces different representations of the same process if they are more convenient or provide deeper insight—different coordinate systems, for example, or classical versus Lagrangian mechanics—even when they make the same predictions.

The datasets we analysed measured the net result of all interactions over the lifetime of the microbial group studied in a mix experiment. Many included several rounds of reproduction, which can compound fitness effects across generations and increase their measured magnitude. Canonical theory might perform better over shorter timescales, but measuring and analysing fitness outcomes for individual cells and virions is unlikely to resolve most issues. We found strong non-additive effects, for example, in datasets measuring the survival of starving *Myxococcus* or *Dictyostelium* cells to become spores in fruiting bodies with little to no reproduction over the course of the experiment. And in many experiments initial frequency determined which strain was most fit—a result that cannot be caused by independent additive effects of self and neighbour genotype. More generally, biologically important interactions among microbes are often difficult to express in terms of single-generation effects on individual fitness. A key benefit of shared external goods like siderophores and virulence toxins, for example, is often that they increase local carrying capacity to allow further rounds of cell growth and division. Because microbes reproduce quickly, their social groups and interactions often span generations.

For many research questions, general social evolution theory may not be 'the right tool for the job'. General theory is best for comparing different systems on the same terms and deriving general principles. For specific biological questions, though, it is often best to use models tailored to the biology of the system [20,45]. Biologically specific models can help researchers understand the often nontrivial fitness effects of microbial phenotypes [42,43,46]. Even fixed differences in reproductive rate or lag time, for example, can have frequency-dependent and non-additive fitness effects when there is local resource competition [47]. Following the time dynamics of mix experiments can be especially informative, as the specificity and richness of time-series data allows strong tests of hypotheses that would otherwise be indistinguishable based on the initial and final time points alone [47]. Analytical results are useful because they can clearly identify all the effects and interactions in a model. But the complexity of biologically specific models often requires numerical solution, making them more difficult to analyse.

### (b) Reformulating theory: challenges and prospects

If we wanted social evolution theory to be more useful for quantitative analysis of microbial data, what does it need to

do? Kin and multilevel selection are statistical descriptions of evolution. Like statistical mechanics in physics, they describe the behaviour of aggregate properties of a system (allele frequency, genotype fitness, etc.) in terms of the probabilistic behaviour of its individual parts. So we can frame the theoretical challenge in statistical terms: How should we specify the statistical model describing social selection? What variables should it include, and in what functional form?

The main problem with canonical kin and multilevel selection theory is their quantitative genetic approach formulated in terms of additive, independent effects on Wrightian fitness—appropriate for macroscopic phenotypes like sex ratio, perhaps, but not for microbial fitness. To be analytically useful for microbes, theory would allow fitness effects spanning orders of magnitude with generically non-linear, genotype-dependent responses. It would also be expressible in terms of both strain and multilevel fitness outcomes, be robust to different types of biological interaction, and allow quantitative comparisons across very different systems.

One approach is to extend existing theory—adding terms to Hamilton's rule to handle non-additive effects, for example [3,39], or including multilevel selection terms for trait interactions, intermediate optima and selection peaks [24,48]. But this approach still requires researchers to translate their *ad hoc* data analyses into the terminology of theory. Another approach is to reformulate theory in terms more easily applied to microbial data. Modelling fitness and frequency effects over logarithmic scales seems especially promising, though it is unclear how to best deal with zeroes that occur in single-strain groups and when a genotype becomes undetectible in the descendant population.

## (c) Conclusion

Mathematical theory can play many roles in biological research. Theory can be a conceptual or heuristic guide. It can be a proof of concept to test the verbal logic of hypotheses [49]. It can also be an analytical tool that provides quantitative insight into the biology of a system—epidemiology's $R_0$, for example. We found that kin and multilevel selection theory are both conceptually useful for microbial data, but they also both have substantial analytical shortcomings. The early development of these theories explicitly considered microbes [50,51], but they have mostly been formulated for metazoan biology. If applying theory developed for one set of taxa to another very different set is a test of that theory's explanatory power, kin and multilevel selection both have room for improvement. Taking a quantitative, data-driven approach can help us move beyond verbal and conceptual arguments to find a productive path where both theories are useful analytical tools for understanding social evolution in all branches of life.

Data accessibility. Data and analysis code for this study are available from the Dryad Digital Repository: https://doi.org/10.5061/dryad.g4f4qrfm5 [55].

Authors' contributions. Conceptualization, Data curation, Formal analysis, Investigation, Methodology, Project administration, Software, Validation, Visualization, Writing-original draft, Writing-review & editing: RFI. Data curation, Funding acquisition, Investigation, Methodology, Project administration, Resources, Supervision, Writing-review & editing. Conceived project: js. Designed analyses: js, RFI. Collected data from published studies: js, RFI. Analysed data: js. Interpreted results and organized their presentation: js, RFI. Wrote manuscript: js, RFI.

Competing interests. We declare we have no competing interests.

Funding. This material is partly based upon work supported by the National Science Foundation under grant nos DEB1204352, IOS1256416 and DEB1146375 to J. E. Strassmann and D. C. Queller (Washington Univ. in St Louis).

Acknowledgements. Many thanks to R. Allen, D. Brock, S. Brown, S. Diggle, L. de Vargas Roditi, A. Eldar, J. Gore, I. Gudelj, W.-D. Hardt, A. Jousset, F. Harrison, R. Kümmerli, A. Luján, B. Raymond, O. Rendueles, A. Ross-Gillespie, M. Vasse, G. J. Velicer, J. Wolf and E. Yurtsev for generously sharing data. Thanks to A. Traulsen for discussion and comments on the manuscript.

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
