## [Peer Review File · Proceedings of the Royal Society B: Biological Sciences]

Review History

RSPB-2020-0364.R0 (Original submission)

Review form: Reviewer 1

Recommendation

Accept with minor revision (please list in comments)

Scientific importance: Is the manuscript an original and important contribution to its field?

Good

General interest: Is the paper of sufficient general interest?

Good

Quality of the paper: Is the overall quality of the paper suitable?

Good

Is the length of the paper justified?

Yes

Should the paper be seen by a specialist statistical reviewer?

No

Do you have any concerns about statistical analyses in this paper? If so, please specify them explicitly in your report.

No

It is a condition of publication that authors make their supporting data, code and materials available - either as supplementary material or hosted in an external repository. Please rate, if applicable, the supporting data on the following criteria.

Is it accessible?

Yes

Is it clear?

Yes

Is it adequate?

Yes

Do you have any ethical concerns with this paper?

No

Comments to the Author

This is an interesting paper, investigating the value of using canonical social evolution theory in microbial systems. The key conclusions are likely to be fairly unsurprising for theoreticians working in this field, but these points may be less obvious to empiricists – particularly microbiologist. In that sense, I wonder if it might be useful to explain the canonical theory in a bit more detail, to make them more accessible to this target audience? From line 80.

I like the assessment of the importance of strong selection and non-additive effects in existing experiments and that models that include these perform better. It might be useful to briefly spell out why only strong selection and non-additive effects are likely the key differences with respect canonical theory in the introduction. This will make the results flow better

I also think it would be worthwhile to spell out the value of analytical results versus simulations when talking about context-specific models (Line 238)

Overall, a really nice paper. A nice distraction from viruses and Microsoft Teams! Thanks!

Review form: Reviewer 2

Recommendation

Reject – article is scientifically unsound

Scientific importance: Is the manuscript an original and important contribution to its field?

Acceptable

General interest: Is the paper of sufficient general interest?

Excellent

Quality of the paper: Is the overall quality of the paper suitable?

Acceptable

Is the length of the paper justified?

Yes

Should the paper be seen by a specialist statistical reviewer?

Yes

Do you have any concerns about statistical analyses in this paper? If so, please specify them explicitly in your report.

No

It is a condition of publication that authors make their supporting data, code and materials available - either as supplementary material or hosted in an external repository. Please rate, if applicable, the supporting data on the following criteria.

Is it accessible?

Yes

Is it clear?

Yes

Is it adequate?

Yes

Do you have any ethical concerns with this paper?

No

Comments to the Author

In Evaluating kin and group selection as tools for quantitative analysis of microbial data, the authors compare the goodness of fit of different statistical models for various fitness measures from previously published microbial mixing experiments. They use this analysis to argue that statistical models based on kin and group selection theory provide poor fit to experimental data due to strong selection and non-additive effects of social traits.

The broader question of this manuscript, which could be phrased as how to homogenize the analyses of experimental social evolution from multiple systems, is definitely interesting and timely. In addition, the manuscript is well-written and the methods well-presented. However, there are some important conceptual issues which have made it difficult for me to understand the value of the analyses made by the authors.

First, I find the reference to kin and group selection theory very misleading as none of the experiments described concern populations with structure that is relevant for these frameworks (i.e., such that individuals are more likely to interact either with relatives or within groups). Individuals within the populations studied interact at random. In terms of kin and group selection theory, this means that relatedness is zero and that there is no competition between groups, respectively.

So why call upon notions of kin and group selection when such selection pressures are absent? If experimental populations are well mixed, we are rather looking at classical frequency dependent selection dynamics in randomly interacting and competing population (with the complication of demographic changes).

If, however, the experimental populations do in fact exhibit structure, for instance due to limited dispersal, then the analysis performed by the authors is misguided as the dynamics (and thus fitness measures) of different strains should depend on indirect fitness effects and relatedness, which is not investigated.

The notion of kin selection also makes the comparison with metazoans confusing (l. 24, 209, 270). None of the studies cited l. 24 estimate individual fitness as in the present manuscript but rather are based on estimates of relatedness.

In addition, while it is true that many empirical studies have used the framework of kin selection to make qualitative predictions about social traits among relatives, the only real success with respect to quantitative predictions being borne out in natural populations concerns sex ratio. I would therefore tone down the multiple mentions that kin selection has been devised and successful to explain quantitative social traits in metazoans.

Another problem with the methods used in this manuscript is that they are based on Price's equation (Appendix 1), which works when considering at most one round of reproduction. Notions of fitness from this framework are therefore relevant over one round of reproduction (e.g., number of successful offspring of an individual). However, the authors define fitness from experimental data as the ratio of final to initial number of individuals, presumably encompassing many rounds of reproduction. For multiple rounds of reproduction, one can extend Price's equation to show that the ratio of final to initial number of individuals actually corresponds to the geometric (rather than arithmetic) mean of individual fitness over all reproduction events. Therefore the author's assumption that even in the simplest additive case, theory should lead one to expect that the ratio of final to initial number of individuals is a linear function of genotype and frequency is erroneous. Perhaps some of the non-additivity or non-multiplicativity effects that are found are explained by this, rather than some special nature of social behavior in microbes.

To conclude, while the manuscript tackles an interesting question, the framing of the question in terms of kin and/or group selection is confusing and unnecessary. This frame of reference impedes from studying relevant fitness measures for social evolution in populations that presumably do not exhibit kin or group structure. In addition, the authors theoretical basis (from Price's equation) is not suited to investigate the multiple reproduction rounds that characterize the microbial populations they wish to understand. It is perhaps true that social behaviors in microbes have non-additive effects (and this would definitely be interesting to describe), but the authors have not made a convincing case for it. With this in mind, it seems to me that the study requires considerable changes to reach publication.

Decision letter (RSPB-2020-0364.R0)

27-Apr-2020

Dear Dr Smith:

I am writing to inform you that your manuscript RSPB-2020-0364 entitled "Evaluating kin and group selection as tools for quantitative analysis of microbial data" has, in its current form, been rejected for publication in Proceedings B.

This action has been taken on the advice of referees, who have recommended that substantial revisions are necessary. With this in mind we would be happy to consider a resubmission, provided the comments of the referees are fully addressed. However please note that this is not a provisional acceptance.

The resubmission will be treated as a new manuscript. However, we will approach the same reviewers if they are available and it is deemed appropriate to do so by the Editor. Please note that resubmissions must be submitted within six months of the date of this email. In exceptional

circumstances, extensions may be possible if agreed with the Editorial Office. Manuscripts submitted after this date will be automatically rejected.

Sincerely,
 Professor Hans Heesterbeek
 mailto: proceedingsb@royalsociety.org

Associate Editor

Board Member: 1

Comments to Author:

I have now received two referee reports and as you can see they have different views of the paper. This reflects in part that one is an empiricist wedded in theory and the second referee is a theoretician. I would like you to reply in detail to the comments of both referees. I agree that a introduction for microbiologists to the theory as suggested by referee 1 would be very useful - this is a field where lots of people really don't understand the theory they are saying that they are testing and there is a lot of confusion. I think you can make a real impact by adding this here.

Referee 2 is highly critical. My sense is that there are some important points here, but that you may well be able to clarify some key issues in a rewrite and by directly addressing the comments.

The reason you wrote this paper is that there is a lot of confusion about the application of this theory and as such you need to address these issues. I would therefore like to see a revised manuscript with a detailed reply that clarifies and also makes clear arguments where you disagree with the referee.

Reviewer(s)' Comments to Author:

Referee: 1

Comments to the Author(s)

This is an interesting paper, investigating the value of using canonical social evolution theory in microbial systems. The key conclusions are likely to be fairly unsurprising for theoreticians working in this field, but these points may be less obvious to empiricists - particularly microbiologist. In that sense, I wonder if it might be useful to explain the canonical theory in a bit more detail, to make them more accessible to this target audience? From line 80.

I like the assessment of the importance of strong selection and non-additive effects in existing experiments and that models that include these perform better. It might be useful to briefly spell out why only strong selection and non-additive effects are likely the key differences with respect canonical theory in the introduction. This will make the results flow better

I also think it would be worthwhile to spell out the value of analytical results versus simulations when talking about context-specific models (Line 238)

Overall, a really nice paper. A nice distraction from viruses and Microsoft Teams! Thanks!

Referee: 2

Comments to the Author(s)

In Evaluating kin and group selection as tools for quantitative analysis of microbial data, the authors compare the goodness of fit of different statistical models for various fitness measures from previously published microbial mixing experiments. They use this analysis to argue that statistical models based on kin and group selection theory provide poor fit to experimental data due to strong selection and non-additive effects of social traits.

The broader question of this manuscript, which could be phrased as how to homogenize the analyses of experimental social evolution from multiple systems, is definitely interesting and timely. In addition, the manuscript is well-written and the methods well-presented. However, there are some important conceptual issues which have made it difficult for me to understand the value of the analyses made by the authors.

First, I find the reference to kin and group selection theory very misleading as none of the experiments described concern populations with structure that is relevant for these frameworks (i.e., such that individuals are more likely to interact either with relatives or within groups). Individuals within the populations studied interact at random. In terms of kin and group selection theory, this means that relatedness is zero and that there is no competition between groups, respectively.

So why call upon notions of kin and group selection when such selection pressures are absent? If experimental populations are well mixed, we are rather looking at classical frequency dependent selection dynamics in randomly interacting and competing population (with the complication of demographic changes).

If, however, the experimental populations do in fact exhibit structure, for instance due to limited dispersal, then the analysis performed by the authors is misguided as the dynamics (and thus fitness measures) of different strains should depend on indirect fitness effects and relatedness, which is not investigated.

The notion of kin selection also makes the comparison with metazoans confusing (l. 24, 209, 270). None of the studies cited l. 24 estimate individual fitness as in the present manuscript but rather are based on estimates of relatedness.

In addition, while it is true that many empirical studies have used the framework of kin selection to make qualitative predictions about social traits among relatives, the only real success with respect to quantitative predictions being borne out in natural populations concerns sex ratio. I would therefore tone down the multiple mentions that kin selection has been devised and successful to explain quantitative social traits in metazoans.

Another problem with the methods used in this manuscript is that they are based on Price's equation (Appendix 1), which works when considering at most one round of reproduction. Notions of fitness from this framework are therefore relevant over one round of reproduction (e.g., number of successful offspring of an individual). However, the authors define fitness from experimental data as the ratio of final to initial number of individuals, presumably encompassing many rounds of reproduction. For multiple rounds of reproduction, one can extend Price's equation to show that the ratio of final to initial number of individuals actually corresponds to the geometric (rather than arithmetic) mean of individual fitness over all reproduction events. Therefore the author's assumption that even in the simplest additive case, theory should lead one

to expect that the ratio of final to initial number of individuals is a linear function of genotype and frequency is erroneous. Perhaps some of the non-additivity or non-multiplicativity effects that are found are explained by this, rather than some special nature of social behavior in microbes.

To conclude, while the manuscript tackles an interesting question, the framing of the question in terms of kin and/or group selection is confusing and unnecessary. This frame of reference impedes from studying relevant fitness measures for social evolution in populations that presumably do not exhibit kin or group structure. In addition, the authors theoretical basis (from Price's equation) is not suited to investigate the multiple reproduction rounds that characterize the microbial populations they wish to understand. It is perhaps true that social behaviors in microbes have non-additive effects (and this would definitely be interesting to describe), but the authors have not made a convincing case for it. With this in mind, it seems to me that the study requires considerable changes to reach publication.

Author's Response to Decision Letter for (RSPB-2020-0364.R0)

See Appendix A.

RSPB-2020-1657.R0

Review form: Reviewer 2

Recommendation

Accept with minor revision (please list in comments)

Scientific importance: Is the manuscript an original and important contribution to its field?

Good

General interest: Is the paper of sufficient general interest?

Good

Quality of the paper: Is the overall quality of the paper suitable?

Good

Is the length of the paper justified?

Yes

Should the paper be seen by a specialist statistical reviewer?

Yes

Do you have any concerns about statistical analyses in this paper? If so, please specify them explicitly in your report.

No

It is a condition of publication that authors make their supporting data, code and materials available - either as supplementary material or hosted in an external repository. Please rate, if applicable, the supporting data on the following criteria.

Is it accessible?

Yes

Is it clear?

Yes

Is it adequate?

Yes

Do you have any ethical concerns with this paper?

No

Comments to the Author

I thank the authors for their clear and detailed response to my earlier comments. With their changes, I see much better the manuscript's intentions. Coming from the theory side of things, the manuscript has helped me understand how microbiologists study and think about social evolution. My only remaining comment is that I think it would be helpful for readers to discuss in a bit more depth what their results can say about the nature of non-additive fitness effects, either in the results section (l. 154-172) or in the discussion, especially in the context of measuring fitness as the ratio of final to initial abundance. For instance, wouldn't linear effects on "individual" fitness (e.g., doubling time, R_0 or per capita growth rate) generate non-linear effects on fitness as the ratio of final to initial abundance?

Decision letter (RSPB-2020-1657.R0)

10-Aug-2020

Dear Dr smith:

Your manuscript has now been peer reviewed and the reviews have been assessed by an Associate Editor. The reviewer's comments (not including confidential comments to the Editor) are included at the end of this email for your reference. As you will see, the reviewer has some specific requests for your manuscript and we would like to invite you to revise your manuscript to address them.

Research ethics:

Use of animals and field studies:

It is a condition of publication that you make available the data and research materials supporting the results in the article (<https://royalsociety.org/journals/authors/author-guidelines/#data>). Datasets should be deposited in an appropriate publicly available repository and details of the associated accession number, link or DOI to the datasets must be included in the Data Accessibility section of the article (<https://royalsociety.org/journals/ethics-policies/data-sharing-mining/>). Reference(s) to datasets should also be included in the reference list of the article with DOIs (where available).

Please submit a copy of your revised paper within three weeks. If we do not hear from you within this time your manuscript will be rejected. If you are unable to meet this deadline please let us know as soon as possible, as we may be able to grant a short extension.

Best wishes,
Professor Hans Heesterbeek
mailto: proceedingsb@royalsociety.org

Reviewer(s)' Comments to Author:
Referee: 2

Comments to the Author(s).

I thank the authors for their clear and detailed response to my earlier comments. With their changes, I see much better the manuscript's intentions. Coming from the theory side of things, the manuscript has helped me understand how microbiologists study and think about social evolution. My only remaining comment is that I think it would be helpful for readers to discuss in a bit more depth what their results can say about the nature of non-additive fitness effects, either in the results section (l. 154-172) or in the discussion, especially in the context of measuring fitness as the ratio of final to initial abundance. For instance, wouldn't linear effects on "individual" fitness (e.g., doubling time, R_0 or per capita growth rate) generate non-linear effects on fitness as the ratio of final to initial abundance?

Author's Response to Decision Letter for (RSPB-2020-1657.R0)

See Appendix B.

RSPB-2020-1657.R1 (Revision)

Review form: Reviewer 3

Recommendation

Major revision is needed (please make suggestions in comments)

Scientific importance: Is the manuscript an original and important contribution to its field?

Good

General interest: Is the paper of sufficient general interest?

Good

Quality of the paper: Is the overall quality of the paper suitable?

Good

Is the length of the paper justified?

No

Should the paper be seen by a specialist statistical reviewer?

Yes

Do you have any concerns about statistical analyses in this paper? If so, please specify them explicitly in your report.

Yes

It is a condition of publication that authors make their supporting data, code and materials available - either as supplementary material or hosted in an external repository. Please rate, if applicable, the supporting data on the following criteria.

Is it accessible?

N/A

Is it clear?

N/A

Is it adequate?

N/A

Do you have any ethical concerns with this paper?

No

Comments to the Author

The question of how social traits evolve in microbes is an important one. Consequently, rigorous evolutionary theory that can be applied to microbial data is a major question. I like the approach of trying out different models on a well-chosen selection of empirical papers. The contribution that the paper makes is the finding that complex models with polynomial and interaction terms and log-transformed fitness yield better fit, suggesting that the fitness consequences of behaviour are more complex than realized. The paradigm of the conflict between canonical and non-linear approaches was not helpful to me. This is more a simple vs. complex model question – see point 1 below. The ‘canonical approach’ is not set in stone – the authors made the decision to use the regression approach designed for a single generation to model longterm selection. They chose the variables to include in the canonical regression and the more complex regression. The paper would be greatly more understandable if they set up their questions more clearly instead of working to convince the reader that they are busting a canon. Essentially I’d like to see this paper rewritten for the non-theorist and non-microbial person, who need much more help in following this paper.

1. I like how this comes together, but the pieces need to be referenced. Lande and Arnold 1983 introduced an approach for analyzing multiple traits with polynomial and interaction terms, which has been applied to multilevel selections – see multiple papers by Goodnight. Game theory is essentially an analysis of how the fitness consequences of behaviour by one player can depend on the behaviour of the other player.
2. Paragraph line 56. This paragraph is indicating that there is a big problem with evolutionary theory, but only identifies the problem obliquely without actually demonstrating that problem. The last sentence “Or perhaps the weak, additive fitness effects common in canonical theory [3, 2, 25, 9, 26] are more problematic in microbial systems where large populations and rapid reproduction make selection especially effective.” bothers me as the “Or” implies an “Either” that is not obvious. It feels like a rhetorical rather than a scientific argument.
3. Line 60. “We focus on a “common mix experiment” design for assaying the fitness effects of social interactions by measuring how local genotype frequency affects performance” This paper would be considerably more understandable if this sentence were unpacked into a few paragraphs – what is a common mix experiment, why local genotype freq is a measure of social environment, how the phenotypic difference between strains is included in the model. The introduction talks about how fitness is assessed in microbial populations, but not how social behaviour can be measured.
4. Fig 1. I get that this is supposed to demonstrate the problem with theory, but it is not

understandable on its own. Microbial mix, Wrightian fitness are not explained, nor are the Kin and multilevel selection models used to create predictions. Theories can't use models, researchers use models. The lines show the predicted fitness (or fitness ratio) from a model, not the model itself. I have not seen the model when I encounter the figure, so I can't even begin to know how astonished I should be by predictions. 1A legend needs to actually say that the white microbes produce a resource (black dots) that both black and white microbes need the resource, and that the fitness is indicated by the number of microbes in the final population (and specify 'final'). If white microbes are extirpated in the 3rd population, why does it have higher fitness than the 4th population?

5. Figure design comments: The contrast between shades of gray for the different lines is hard to identify against ggplot's default gray background. Total population and non-producer are extremely similar, and from the legend, I expect to find these classes in all 3 figure panels. The points are not jittered so we don't see the actual data very well.

6. The authors identify the theory they are evaluating in the methods under "applying theory". I would like to see this in the introduction.

7. "Canonical" is not a clearly defined term – see

<https://en.wikipedia.org/wiki/Canonical> and beyond these meanings there's a confusing array of canonical correlation analysis, canonical correspondence analysis, and more in ecological statistics, and so it isn't immediately clear what the authors mean. I'd recommend using "simple" here.

8. As I read this section, the problematic approach (canonical) resulted from how authors themselves initially conceptualized the theory and applied it to microbes, rather than there being any examples of poor practice in the literature. In particular, G is defined as initial strain frequency, a constant, but in this system, genotype frequency will change rapidly.

9. See Multilevel selection theory and evidence: a critique of Gardner, 2015. C. J. GOODNIGHT. doi: 10.1111/jeb.12685 do= for why not to use a Price equation approach to multilevel selection. The kin selection approach meets Goodnight's approach to multilevel selection.

10. Line 154. Intermediate optima, trait interactions and selection peaks are also 'canonical' – e.g. Wright's fitness landscape.

11. Line 197 Mixing frequency? Same as strain frequency?

12. Fig 2. No metrics are given for these histograms, so we are judging by eye. Give someone unfamiliar with judging models by AIC enough information to evaluate these figures. What would show that 'linear' models are better? Can any statistical tests be done? All the results are figures for the reader to evaluate by eye, without much guidance.

13. Line 275. If all results were done in `lm()`, (calculations and statistics section) they are all linear models and least squares regression. Transformations of variables, polynomials, and interaction are a common part of linear additive models. All the models as described are statistically linear, least squares regressions (see https://en.wikipedia.org/wiki/Polynomial_regression, <https://stats.stackexchange.com/questions/92065/why-is-polynomial-regression-considered-a-special-case-of-multiple-linear-regres>).

14. Line 285. Researchers commonly add a constant to data before log transforming to deal with zeros. Would doing that affect these results?

Decision letter (RSPB-2020-1657.R1)

20-Oct-2020

Dear Dr Smith:

Your manuscript has now been peer reviewed. The reviewer's comments (not including confidential comments to the Editor) are included at the end of this email for your reference. I

would like to point out that this is a new reviewer because the reviewer who provided the comments for your recent revision was unavailable. A new reviewer invariably raises new issues, as is the nature of science. Out of fairness to the time and effort that the reviewer has invested, I cannot accept the manuscript in its current form. Out of fairness to you (and the previous reviewers), however, I also cannot request a very thorough new revision while we were on the road to convergence. I do think that the new reviewer makes a number of very good points that could further improve your manuscript and make it more widely appealing. I urge you therefore to seriously consider and address the suggestions by the reviewer.

We do not allow multiple rounds of revision so we urge you to make every effort to fully address all of the comments at this stage. If deemed necessary by the Associate Editor, your manuscript will be sent back to one or more of the original reviewers for assessment. If the original reviewers are not available we may invite new reviewers, but are unlikely to do so in your particular case. Please note that we cannot guarantee eventual acceptance of your manuscript at this stage.

Research ethics:

Use of animals and field studies:

It is a condition of publication that you make available the data and research materials supporting the results in the article (<https://royalsociety.org/journals/authors/author-guidelines/#data>). Datasets should be deposited in an appropriate publicly available repository and details of the associated accession number, link or DOI to the datasets must be included in the Data Accessibility section of the article (<https://royalsociety.org/journals/ethics-policies/data-sharing-mining/>). Reference(s) to datasets should also be included in the reference list of the article with DOIs (where available).

If you wish to submit your data to Dryad (<http://datadryad.org/>) and have not already done so you can submit your data via this link [http://datadryad.org/submit?journalID=RSPB&manu=\(Document not available\)](http://datadryad.org/submit?journalID=RSPB&manu=(Document+not+available)), which will take you to your unique entry in the Dryad repository.

Please submit a copy of your revised paper within three weeks. If we do not hear from you within this time your manuscript will be rejected. If you are unable to meet this deadline please let us know as soon as possible, as we may be able to grant a short extension.

Best wishes,
Professor Hans Heesterbeek
Editor, Proceedings B
mailto: proceedingsb@royalsociety.org

Reviewer(s)' Comments to Author:

Referee: 3

Comments to the Author(s)

The question of how social traits evolve in microbes is an important one. Consequently, rigorous evolutionary theory that can be applied to microbial data is a major question. I like the approach of trying out different models on a well-chosen selection of empirical papers. The contribution that the paper makes is the finding that complex models with polynomial and interaction terms and log-transformed fitness yield better fit, suggesting that the fitness consequences of behaviour are more complex than realized. The paradigm of the conflict between canonical and non-linear approaches was not helpful to me. This is more a simple vs. complex model question – see point 1 below. The ‘canonical approach’ is not set in stone – the authors made the decision to use the regression approach designed for a single generation to model longterm selection. They chose the variables to include in the canonical regression and the more complex regression. The paper would be greatly more understandable if they set up their questions more clearly instead of working to convince the reader that they are busting a canon. Essentially I’d like to see this paper rewritten for the non-theorist and non-microbial person, who need much more help in following this paper.

1. I like how this comes together, but the pieces need to be referenced. Lande and Arnold 1983 introduced an approach for analyzing multiple traits with polynomial and interaction terms, which has been applied to multilevel selections – see multiple papers by Goodnight. Game theory is essentially an analysis of how the fitness consequences of behaviour by one player can depend on the behaviour of the other player.
2. Paragraph line 56. This paragraph is indicating that there is a big problem with evolutionary theory, but only identifies the problem obliquely without actually demonstrating that problem. The last sentence “Or perhaps the weak, additive fitness effects common in canonical theory [3, 2, 25, 9, 26] are more problematic in microbial systems where large populations and rapid reproduction make selection especially effective.” bothers me as the “Or” implies an “Either” that is not obvious. It feels like a rhetorical rather than a scientific argument.
3. Line 60. “We focus on a “common mix experiment” design for assaying the fitness effects of social interactions by measuring how local genotype frequency affects performance” This paper would be considerably more understandable if this sentence were unpacked into a few paragraphs – what is a common mix experiment, why local genotype freq is a measure of social environment, how the phenotypic difference between strains is included in the model. The introduction talks about how fitness is assessed in microbial populations, but not how social behaviour can be measured.
4. Fig 1. I get that this is supposed to demonstrate the problem with theory, but it is not understandable on its own. Microbial mix, Wrightian fitness are not explained, nor are the Kin and multilevel selection models used to create predictions. Theories can’t use models, researchers use models. The lines show the predicted fitness (or fitness ratio) from a model, not the model itself. I have not seen the model when I encounter the figure, so I can’t even begin to know how astonished I should be by predictions. 1A legend needs to actually say that the white microbes produce a resource (black dots) that both black and white microbes need the resource, and that the fitness is indicated by the number of microbes in the final population (and specify ‘final’). If white microbes are extirpated in the 3rd population, why does it have higher fitness than the 4th population?
5. Figure design comments: The contrast between shades of gray for the different lines is hard to identify against ggplot’s default gray background. Total population and non-producer are extremely similar, and from the legend, I expect to find these classes in all 3 figure panels. The points are not jittered so we don’t see the actual data very well.
6. The authors identify the theory they are evaluating in the methods under “applying theory”. I would like to see this in the introduction.
7. “Canonical” is not a clearly defined term – see <https://en.wikipedia.org/wiki/Canonical> and beyond these meanings there’s a confusing array of canonical correlation analysis, canonical correspondence analysis, and more in ecological statistics, and so it isn’t immediately clear what the authors mean. I’d recommend using “simple” here.
8. As I read this section, the problematic approach (canonical) resulted from how authors themselves initially conceptualized the theory and applied it to microbes, rather than there being any examples of poor practice in the literature. In particular, G is defined as initial strain frequency, a constant, but in this system, genotype frequency will change rapidly.
9. See Multilevel selection theory and evidence: a critique of Gardner, 2015. C. J. GOODNIGHT. doi: 10.1111/jeb.12685 do= for why not to use a Price equation approach to multilevel selection. The kin selection approach meets Goodnight’s approach to multilevel selection.
10. Line 154. Intermediate optima, trait interactions and selection peaks are also ‘canonical’ – e.g. Wright’s fitness landscape.
11. Line 197 Mixing frequency? Same as strain frequency?
12. Fig 2. No metrics are given for these histograms, so we are judging by eye. Give someone unfamiliar with judging models by AIC enough information to evaluate these figures. What would show that ‘linear’ models are better? Can any statistical tests be done? All the results are figures for the reader to evaluate by eye, without much guidance.
13. Line 275. If all results were done in $\text{lm}()$, (calculations and statistics section) they are all linear models and least squares regression. Transformations of variables, polynomials, and interaction are a common part of linear additive models. All the models as described are statistically linear,

least squares regressions (see https://en.wikipedia.org/wiki/Polynomial_regression, <https://stats.stackexchange.com/questions/92065/why-is-polynomial-regression-considered-a-special-case-of-multiple-linear-regres>).

14. Line 285. Researchers commonly add a constant to data before log transforming to deal with zeros. Would doing that affect these results?

Author's Response to Decision Letter for (RSPB-2020-1657.R1)

See Appendix C.

RSPB-2020-1657.R2 (Revision)

Review form: Reviewer 3

Recommendation

Accept with minor revision (please list in comments)

Scientific importance: Is the manuscript an original and important contribution to its field?

Excellent

General interest: Is the paper of sufficient general interest?

Good

Quality of the paper: Is the overall quality of the paper suitable?

Acceptable

Is the length of the paper justified?

Yes

Should the paper be seen by a specialist statistical reviewer?

No

Do you have any concerns about statistical analyses in this paper? If so, please specify them explicitly in your report.

No

It is a condition of publication that authors make their supporting data, code and materials available - either as supplementary material or hosted in an external repository. Please rate, if applicable, the supporting data on the following criteria.

Is it accessible?

Yes

Is it clear?

Yes

Is it adequate?

No

Do you have any ethical concerns with this paper?

No

Comments to the Author

Rereview of smith and Inglis

This manuscript reads much better, and the authors have addressed many of my concerns. However, there is considerable inconsistency in terminology that hindered my ability to follow the arguments. The manuscript needs editing for consistency in terminology for initial genotype frequency, kin selection, and components of multilevel selection. The figures need to be remade to use consistent terminology for initial genotype frequency (G) on the X-axis and to be accessible. The authors must indicate which panels indicate kin selection, the group component of multilevel selection, and the within group component of multilevel selection. Explanations of theory should include the basics and indicate how they are specified for analysis of mix experiments.

Evaluation of responses to Reviewer 3.

General point

I will leave the use of canonical (as opposed to data driven) theory up to the editor. 'Canonical' has connotations related to its use as a synonym for orthodoxy, authority and church law that I find overwhelming, but others may find it more useful. I see the appeal of breaking canon, but I have found that theoreticians are aware of the assumptions and consequent limitations of their theory, and seen that models with weak selection and additive effects generate complex nonlinear predictions.

Responses to my comments

1. Satisfactory
2. Satisfactory – this paragraph reads much better
3. Mostly satisfactory. Fig 1A shows that a typical difference in social behaviour is that one genotype is a producer of an external good that both genotypes use and the other genotype is a non-producer. Being a producer/nonproducer is a behaviour that affects the fitness of neighbours. I suggest including the explanation that these traits that affect the fitness of others in the population because it justifies the use of kin selection/multilevel selection.
4. Because many people skim over methods on the first read of a paper, figures should be self-explanatory. Part A is very clear. In part B the legend needs to be more specific. Genotype is ambiguous here, since it could refer to genotype frequency or genotypic value. Suggested wording: B) Estimation of key parameters for kin selection and multilevel selection in a hypothetical dataset with Wrightian fitness and additive effects. The left panel illustrates the analysis of kin selection on the nonproducer strain, with a regression of strain fitness on genotypic value ($g=0$ for producer genotype and $g=1$ for non-producer genotype) and mean group genotype (G = initial frequency of nonproducers) (Table S2, Eqn. S6). The middle panel illustrates the group selection component of multilevel selection, with the regression of total group fitness on mean group genotype G (Table S2, Eqn. S9). The righthand panel describes the regression for the estimated value of the within-group difference in fitness (fitness of non-producers minus the fitness of producers) on mean group genotype G (Table S2, Eqn. S9), the initial frequency of non-producers.
5. While the figures are better, authors can do much more with use of colour, shape of points, size of points, and solid vs. dashed lines to make them accessible to all readers.
6. Ok
7. See initial comments above
8. Acceptable. Though initial genotype frequency does not change over the experiment, the terminology the authors use for it does change throughout the paper, and 'initial' is often dropped.
9. Satisfactory.
10. Satisfactory
11. The clarification is helpful here, but the best protocol is to settle on consistent terminology and use it throughout.
12. Good. Please add to the orientation the equivalent of "Given a set of candidate models

for the data, the preferred model is the one with the minimum AIC value.” (Wikipedia)

13. Good.

14. Ok

Other comments on the manuscript

15. Line 52-53. Weak selection is often assumed in modelling because strong selection changes genetic variance. It is a simplification that often make modelling tractable. Additive variance is the source of parent offspring correlation.

16. In Figure S2, the “fitness models poorly specified for strong selection and nonadditive effects in microbial data.” are linear as expected. In Fig 1, the “lines show fit of canonical fitness models, whose additive fitness effects” show complex curving. Can the authors provide the models for these examples and explain why they portray ‘additive canonical models’ so differently?

17. Line 104-110. Refer to table S2 when defining g and G . How the initial Genotype frequency is also G , the mean group genotype, is explained only here.

18. For the reader’s understanding, label how the equation apply “Malthusian neighbour-modulated fitness (kin selection) with interaction $m \sim g + G + gG$, Malthusian multilevel group selection $M \sim G$, and Malthusian individual selection $\Delta m \sim G$ ”

19. Explain this more clearly “slopes are comparable to intercepts because they measure the total effect of group genotype from zero to one.”

20. Are mixing frequency (line 170) and local genotype frequency (figure 1 legend) the same as initial genotype frequency? Near eqn S13 and figure legend S4 the term mixing proportion is used. Is this initial gene frequency? Figure axes with initial proportion – is this initial genotype frequency? Near eqn 18 mixing frequency is used. Does proportion differ from frequency in these experiments? If so, use the terms “initial genotype frequency” and/or “mean group genotype” consistently throughout the paper, without introducing more synonyms.

21. The finding of frequency-dependent selection for within-fitness differences is unfortunately confusing, standard as that terminology is. It should be made clear here that when the within-group selection on individual trait genotypic value depends on the group trait “initial genotype frequency” and/or “mean group genotype”, that this means there is frequency-dependent selection.

22. I suggest using ‘kin selection’ generally throughout, with ‘neighbour-modulated fitness approach to kin selection’ used only when this distinction is crucial. Right now they are used synonymously and without any apparent distinction being made. The text around the call to Fig.2 refers to kin selection, but figure 2 itself uses neighbour-modulated fitness.

23. Line 200 spelling of multilevel

24. The text below eqn. S6, kin selection is explained in sometimes in general terms and sometimes specifically for the mix experiments.

25. Consider adding an explanatory section of frequency-dependent selection in the supplemental materials.

26. In the supplementary materials section Strain and group fitness outcomes, add the clarification that neighbor-modulated fitness is used to refer to a particular version of kin selection as in line 103.

27. Table S3 needs final figure references

28. Figure S2 left panel. I can’t distinguish any producer symbols with the dark grey and medium grey on light grey with considerable overlap. With supplementary figures with coloured symbols, the small size of the symbols makes it more difficult to distinguish their colour. There are online tools to help produce figures with accessible contrast.

29. Overall the documentation and organization is clear for the data and code. However, illustrate-issues does not run for the data example. I tried to see if I could answer point 16 myself. However, the code does not run because a chunk of variable creation code was omitted. The sample data set only contains variables strain.B strain.A, initial.number.B, initial.number.A, final.number.B, final.number.A. The next line asks for a subset with multiple variables not yet created, including initial.ratio.A, initial.proportion.A, fitness.A, fitness.B, fitness.total, fitness.ratio.A. Neither initial.ratio.A nor fitness.ratio.A were created in the previous hypothetical example. There are also some apparent name changes e.g. fitness.total.group vs. fitness.

Decision letter (RSPB-2020-1657.R2)

09-Apr-2021

Dear Dr Smith

I am pleased to inform you that your manuscript RSPB-2020-1657.R2 entitled "Evaluating kin and group selection as tools for quantitative analysis of microbial data" has been accepted for publication in Proceedings B, subject to final editing.

The referee has recommended publication, but also suggest a substantial number of minor but important revisions to your manuscript. Therefore, I invite you to respond to the referee's comments and revise your manuscript. Because the schedule for publication is very tight, it is a condition of publication that you submit the revised version of your manuscript within 7 days. If you do not think you will be able to meet this date please let us know.

Sincerely,
 Professor Hans Heesterbeek
 Editor, Proceedings B
<mailto:proceedingsb@royalsociety.org>

Reviewer(s)' Comments to Author:

Referee: 3

Comments to the Author(s)

Rereview of smith and Inglis

This manuscript reads much better, and the authors have addressed many of my concerns. However, there is considerable inconsistency in terminology that hindered my ability to follow the arguments. The manuscript needs editing for consistency in terminology for initial genotype frequency, kin selection, and components of multilevel selection. The figures need to be remade to use consistent terminology for initial genotype frequency (G) on the X-axis and to be accessible. The authors must indicate which panels indicate kin selection, the group component of multilevel selection, and the within group component of multilevel selection. Explanations of theory should include the basics and indicate how they are specified for analysis of mix experiments.

Evaluation of responses to Reviewer 3.

General point

I will leave the use of canonical (as opposed to data driven) theory up to the editor. 'Canonical' has connotations related to its use as a synonym for orthodoxy, authority and church law that I find overwhelming, but others may find it more useful. I see the appeal of breaking canon, but I

have found that theoreticians are aware of the assumptions and consequent limitations of their theory, and seen that models with weak selection and additive effects generate complex nonlinear predictions.

Responses to my comments

1. Satisfactory

2. Satisfactory – this paragraph reads much better

3. Mostly satisfactory. Fig 1A shows that a typical difference in social behaviour is that one genotype is a producer of an external good that both genotypes use and the other genotype is a non-producer. Being a producer/nonproducer is a behaviour that affects the fitness of neighbours. I suggest including the explanation that these traits that affect the fitness of others in the population because it justifies the use of kin selection/multilevel selection.

4. Because many people skim over methods on the first read of a paper, figures should be self-explanatory. Part A is very clear. In part B the legend needs to be more specific. Genotype is ambiguous here, since it could refer to genotype frequency or genotypic value. Suggested wording: B) Estimation of key parameters for kin selection and multilevel selection in a hypothetical dataset with Wrightian fitness and additive effects. The left panel illustrates the analysis of kin selection on the nonproducer strain, with a regression of strain fitness on genotypic value ($g=0$ for producer genotype and $g=1$ for non-producer genotype) and mean group genotype ($G =$ initial frequency of nonproducers) (Table S2, Eqn. S6). The middle panel illustrates the group selection component of multilevel selection, with the regression of total group fitness on mean group genotype G (Table S2, Eqn. S9). The righthand panel describes the regression for the estimated value of the within-group difference in fitness (fitness of non-producers minus the fitness of producers) on mean group genotype G (Table S2, Eqn. S9), the initial frequency of non-producers.

5. While the figures are better, authors can do much more with use of colour, shape of points, size of points, and solid vs. dashed lines to make them accessible to all readers.

6. Ok

7. See initial comments above

8. Acceptable. Though initial genotype frequency does not change over the experiment, the terminology the authors use for it does change throughout the paper, and 'initial' is often dropped.

9. Satisfactory.

10. Satisfactory

11. The clarification is helpful here, but the best protocol is to settle on consistent terminology and use it throughout.

12. Good. Please add to the orientation the equivalent of "Given a set of candidate models for the data, the preferred model is the one with the minimum AIC value." (Wikipedia)

13. Good.

14. Ok

Other comments on the manuscript

15. Line 52-53. Weak selection is often assumed in modelling because strong selection changes genetic variance. It is a simplification that often make modelling tractable. Additive variance is the source of parent offspring correlation.

16. In Figure S2, the "fitness models poorly specified for strong selection and nonadditive effects in microbial data." are linear as expected. In Fig 1, the "lines show fit of canonical fitness models, whose additive fitness effects" show complex curving. Can the authors provide the models for these examples and explain why they portray 'additive canonical models' so differently?

17. Line 104-110. Refer to table S2 when defining g and G . How the initial Genotype frequency is also G , the mean group genotype, is explained only here.

18. For the reader's understanding, label how the equation apply "Malthusian neighbour-modulated fitness (kin selection) with interaction $m \sim g+G+gG$, Malthusian multilevel group selection $M \sim G$, and Malthusian individual selection $\Delta m \sim G$ "

19. Explain this more clearly "slopes are comparable to intercepts because they measure the total effect of group genotype from zero to one."

20. Are mixing frequency (line 170) and local genotype frequency (figure 1 legend) the same as initial genotype frequency? Near eqn S13 and figure legend S4 the term mixing proportion is

used. Is this initial gene frequency? Figure axes with initial proportion – is this initial genotype frequency? Near eqn 18 mixing frequency is used. Does proportion differ from frequency in these experiments? If so, use the terms “initial genotype frequency” and/or “mean group genotype” consistently throughout the paper, without introducing more synonyms.

21. The finding of frequency-dependent selection for within-fitness differences is unfortunately confusing, standard as that terminology is. It should be made clear here that when the within-group selection on individual trait genotypic value depends on the group trait “initial genotype frequency” and/or “mean group genotype”, that this means there is frequency-dependent selection.

22. I suggest using ‘kin selection’ generally throughout, with ‘neighbour-modulated fitness approach to kin selection’ used only when this distinction is crucial. Right now they are used synonymously and without any apparent distinction being made. The text around the call to Fig.2 refers to kin selection, but figure 2 itself uses neighbour-modulated fitness.

23. Line 200 spelling of multilevel

24. The text below eqn. S6, kin selection is explained in sometimes in general terms and sometimes specifically for the mix experiments.

25. Consider adding an explanatory section of frequency-dependent selection in the supplemental materials.

26. In the supplementary materials section Strain and group fitness outcomes, add the clarification that neighbor-modulated fitness is used to refer to a particular version of kin selection as in line 103.

27. Table S3 needs final figure references

28. Figure S2 left panel. I can’t distinguish any producer symbols with the dark grey and medium grey on light grey with considerable overlap. With supplementary figures with coloured symbols, the small size of the symbols makes it more difficult to distinguish their colour. There are online tools to help produce figures with accessible contrast.

29. Overall the documentation and organization is clear for the data and code. However, illustrate-issues does not run for the data example. I tried to see if I could answer point 16 myself. However, the code does not run because a chunk of variable creation code was omitted. The sample data set only contains variables strain.B strain.A, initial.number.B, initial.number.A, final.number.B, final.number.A. The next line asks for a subset with multiple variables not yet created, including initial.ratio.A, initial.proportion.A, fitness.A, fitness.B, fitness.total, fitness.ratio.A. Neither initial.ratio.A nor fitness.ratio.A were created in the previous hypothetical example. There are also some apparent name changes e.g. fitness.total.group vs. fitness.

Author's Response to Decision Letter for (RSPB-2020-1657.R2)

See Appendix D.

Decision letter (RSPB-2020-1657.R3)

22-Apr-2021

Dear Dr Smith

I am pleased to inform you that your manuscript entitled "Evaluating kin and group selection as tools for quantitative analysis of microbial data" has been accepted for publication in Proceedings B.

You can expect to receive a proof of your article from our Production office in due course, please check your spam filter if you do not receive it. PLEASE NOTE: you will be given the exact page

length of your paper which may be different from the estimation from Editorial and you may be asked to reduce your paper if it goes over the 10 page limit.

Data Accessibility section

Open Access

Paper charges

Sincerely,

Appendix A

Manuscript ID RSPB-2020-0364

Response to reviewers

We thank the reviewers for their comments on our manuscript, “Evaluating kin and group selection as tools for quantitative analysis of microbial data” (*Proceedings B* manuscript ID RSPB-2020-0364). Addressing their concerns with this revision has improved the paper. We hope our findings are now more clear to a diverse audience of empiricists and theoreticians. The attached version indicates the specific changes with **red text**.

Reviewer 1

This is an interesting paper, investigating the value of using canonical social evolution theory in microbial systems. The key conclusions are likely to be fairly unsurprising for theoreticians working in this field, but these points may be less obvious to empiricists—particularly microbiologists. In that sense, I wonder if it might be useful to explain the canonical theory in a bit more detail, to make them more accessible to this target audience? From line 80.

In the Methods we now include an introduction to theory that we hope is more accessible to microbiologists (lines 87–107).

I like the assessment of the importance of strong selection and non-additive effects in existing experiments and that models that include these perform better. It might be useful to briefly spell out why only strong selection and non-additive effects are likely the key differences with respect canonical theory in the introduction. This will make the results flow better.

In the Introduction we now suggest that strong selection and nonadditive effects may be more prevalent in microbial systems because their large populations and rapid reproduction make selection especially effective (line 56).

I also think it would be worthwhile to spell out the value of analytical results versus simulations when talking about context-specific models (Line 238).

In the Discussion we now describe how analytical results, when obtainable, can clearly identify all the effects and interactions in a model. The complexity of biologically-specific models often requires numerical solution, though, making them more difficult to analyze (line 262).

Reviewer 2

...[T]here are some important conceptual issues which have made it difficult for me to understand the value of the analyses made by the authors.

First, I find the reference to kin and group selection theory very misleading as none of the experiments described concern populations with structure that is relevant for these frameworks (*i.e.*, such that individuals are more likely to interact either with relatives or within groups). Individuals within the populations studied interact at random. In terms of kin and group selection theory, this means that relatedness is zero and that there is no competition between groups, respectively.

So why call upon notions of kin and group selection when such selection pressures are absent? If experimental populations are well mixed, we are rather looking at classical frequency dependent selection dynamics in randomly interacting and competing population (with the complication of demographic changes).

If, however, the experimental populations do in fact exhibit structure, for instance due to limited dispersal, then the analysis performed by the authors is misguided as the dynamics (and thus fitness measures) of different strains should depend on indirect fitness effects and relatedness, which is not investigated.

The experiments we analyze are indeed initially well-mixed, and their results can indeed be conceptualized in terms of frequency-dependent selection. We focus on the kin/group selection perspective because that is very frequently the lense through which empirical microbiologists interpret these experiments and the reason they conduct them. Mix experiments are a common method for assaying the fitness effects of social interactions. In practice, they examine how selection acts on a local subpopulation during one part of a microbe's lifecycle—within a single infected host, for example. The results can then be used to assess how selection in a larger metapopulation would be affected by kin/group structure at this scale (non-random distribution of initial genotype frequencies among groups). We now explain this in the revised manuscript (line 60).

The notion of kin selection also makes the comparison with metazoans confusing (*l.* 24, 209, 270). None of the studies cited *l.* 24 estimate individual fitness as in the present manuscript but rather are based on estimates of relatedness.

We agree that it can be confusing how to properly apply kin selection methods to microbial species. That's part of the reason for this study—to make quantitatively explicit what terms in theory actually correspond to in microbiological practice.

The point of the Introduction's first paragraph is to describe how kin selection theory has been useful and influential for empirical studies of social behavior in animals. That motivates why we'd want to evaluate how useful it is for empirical studies of social behavior in microbes. The paragraph also describes some of the methodologies used to study multicellular organisms, which do indeed differ from those for microbes. That's the point—multicellular and microbial studies frequently use different empirical methods, so theory useful for analyzing data from one taxa might not be as useful for the other. None of the arguments we make require that the metazoan studies we cite estimate individual fitness.

In addition, while it is true that many empirical studies have used the framework of kin selection to make qualitative predictions about social traits among relatives, the only real success with respect to quantitative predictions being borne out in natural populations concerns sex ratio. I would therefore tone down the multiple mentions that kin selection has been devised and successful to explain quantitative social traits in metazoans.

We agree that sex ratio is the phenotype for which kin selection theory has been most quantitatively successful, but the manuscript doesn't actually claim that kin selection has made successful quantitative predictions about traits other than sex ratio.

The first paragraph of the Introduction says that kin selection theory is canonically formulated in terms of quantitative-genetic traits (*i.e.* continuous phenotypes affected by many loci with weak effects) and that it's been successfully used to make predictions using ESS models that were tested with comparative data. It may be that most of those predictions were qualitative. In the Discussion we already mention that metazoan predictions are often qualitative (line 224). One might argue that the studies we cite aren't sufficiently explanatory because they don't predict the quantitative values of the quantitative-genetic traits they're studying. But they are considered successful applications of kin selection theory by many in the field, and critiquing the theory's explanatory power in metazoans is beyond the scope of this manuscript.

Lines 24 and 224 are the only times the manuscript mentions the successful application of kin selection theory to metazoans.

Another problem with the methods used in this manuscript is that they are based on Price's equation (Appendix 1), which works when considering at most one round of reproduction. Notions of fitness from this framework are therefore relevant over one round of reproduction (*e.g.*, number of successful offspring of an individual). However, the authors define fitness from experimental data as the ratio of final to initial number of individuals, presumably encompassing many rounds of reproduction. For multiple rounds of reproduction, one can extend Price's equation to show that the ratio of final to initial number of individuals actually corresponds to the geometric (rather than arithmetic) mean of individual fitness over all reproduction events. Therefore the author's assumption that even in the simplest additive case, theory should lead one to expect that the ratio of final to initial number of individuals is a linear function of genotype and frequency is erroneous. Perhaps some of the non-additivity or non-multiplicativity effects that are found are explained by this, rather than some special nature of social behavior in microbes.

One of the difficulties when trying to apply canonical social evolution theory is that the quantity used for individual fitness is often not appropriate for microbial biology. In canonical theory, fitness is usually measured as the number of successful offspring produced by an individual over their lifespan, measured for a discrete, given generation. For microbes, though, the most important component of fitness is often generation time. The lifetime fitness of a dividing *E. coli* cell is always two, for example, and what matters is how fast it divides. You can model selection in continuous time, but that runs into the next problem: many biologically important interactions are difficult to express as fitness effects on individual cells or virions within their own lifetime.

Because microbes reproduce quickly, their interactions often span generations. A key benefit of shared external goods like siderophores and virulence toxins, for example, is often that they increase local carrying capacity and allow further rounds of cell growth and division. So the

biological phenomena we're trying to understand often aren't captured by fitness measures like lifetime number of surviving offspring or instantaneous birth and death rate.

One thing we can do instead is measure the net effect of all interactions that happen over the lifetime of the discrete groups we're studying, and then use that quantity in discrete-time theory. That is, measure fitness in terms of how individuals in the initial population contribute to the final population, potentially including descendants from multiple rounds of reproduction. Measuring net fitness over the course of a mix experiment gives us a performance measure that is relevant to microbial biology.

Whole-experiment Wrightian fitness ($w = n'/n$) is mathematically consistent with the Price equation's description of evolutionary change due to selection. The Price equation is a mathematical identity that relates an ancestral population to a descendant population. And because it's a mathematical identity, it's always true by definition. So the equation "works" for whole-experiment fitness—provided it's interpreted in those terms, as well. In our case, that means we're talking about nonadditive and nonmultiplicative effects on whole-experiment fitness, potentially including multiple rounds of reproduction.

It is true that when there are multiple rounds of reproduction, whole-experiment fitness isn't equal to the fitness of individual cells or virions over their own lifespan. But that's fine because we're not trying to draw conclusions about individual lifetime fitness if that's not what the mix experiments measure. So we're not actually assuming that a linear relationship between initial genotype frequency and n'/n indicates additive effects on individual lifetime fitness—only that it indicates additive effects on n'/n .

Our previous manuscript didn't directly address these issues. They're somewhat underappreciated in general, so when we say that nonadditive fitness effects are widespread in microbes we understand how a reader might assume we mean the lifetime fitness of individual organisms. In the revised manuscript we now discuss the issues more fully (lines 87, 89, 241, 244, Suppl. Math "Fitness measures") and use language that we hope better clarifies our intended meaning (lines 6, 96, 131, 138, 210, 228, Suppl. Math "Fitness measures").

We should note, however, that some of our results do point to nonadditive effects on individual, single-generation fitness. We found nonadditive effects, for example, in *Dictyostelium* and *Myxococcus* experiments where n'/n measured the survival of starving cells to become spores in fruiting bodies with little to no reproduction. And in many experiments initial frequency determined which strain was most fit, a result that cannot be caused by independent additive effects of individual and neighbor genotype on lifetime absolute fitness, even when they're compounded over multiple generations. We've made these results clearer in the revised manuscript (lines 146, 171, 244).

To conclude, while the manuscript tackles an interesting question, the framing of the question in terms of kin and/or group selection is confusing and unnecessary. This frame of reference impedes from studying relevant fitness measures for social evolution in populations that presumably do not exhibit kin or group structure. In addition, the authors' theoretical basis (from Price's equation) is not suited to investigate the multiple reproduction rounds that characterize the microbial populations they wish to understand. It is perhaps true that social behaviors in microbes have non-additive effects (and this would definitely be interesting to describe), but the authors have not made a convincing case for it. With this in mind, it seems to me that the study requires considerable changes to reach publication.

We're not trying to show that microbial social behaviors have nonadditive effects on the lifetime fitness of individual cells or virions. If that were our goal, we agree that we'd need an analytical technique that properly accounts for experiments that included multiple rounds of reproduction.

Instead, our goal is to evaluate prominent theoretical approaches in social evolution as tools for quantitatively analyzing the data that microbiologists collect and consider biologically meaningful. Asking questions about kin and group selection theory without framing those questions in terms of kin and group selection theory would be nonsensical. Our results show that these theories are better analytical tools when they allow for strong nonadditive effects on the fitness measures relevant to microbial biology—the net performance of strains over the entire course of a mix experiment, potentially including multiple rounds of reproduction.

We hope the changes we've made in the revised manuscript better clarify our intended findings.

Appendix B

Manuscript ID RSPB-2020-1657

Response to reviewers

We thank the reviewer for their comments on our manuscript, “Evaluating kin and group selection as tools for quantitative analysis of microbial data” (*Proceedings B* manuscript ID RSPB-2020-1657). We’ve adopted their suggestion to discuss in more detail the biological causes of nonadditive fitness effects. The attached version indicates the specific changes with **red text**.

Reviewer 2

I thank the authors for their clear and detailed response to my earlier comments. With their changes, I see much better the manuscript’s intentions. Coming from the theory side of things, the manuscript has helped me understand how microbiologists study and think about social evolution. My only remaining comment is that I think it would be helpful for readers to discuss in a bit more depth what their results can say about the nature of non-additive fitness effects, either in the results section (l. 154-172) or in the discussion, especially in the context of measuring fitness as the ratio of final to initial abundance. For instance, wouldn’t linear effects on “individual” fitness (*e.g.*, doubling time, R_0 or per capita growth rate) generate non-linear effects on fitness as the ratio of final to initial abundance?

Yes, it can. Simple phenotypes can have nontrivial effects on microbial fitness. That’s one of the reasons why we recommend biologically-specific models. Even a fixed difference in reproductive rate, for example, can create nonadditive and frequency-dependent effects on $w = n'/n$ and w_A/w_B when there’s local resource competition. We now discuss this in the manuscript (line 258).

Appendix C

Manuscript ID RSPB-2020-1657

Response to reviewers

We thank the reviewer for their comments on our manuscript, “Evaluating kin and group selection as tools for quantitative analysis of microbial data” (*Proceedings B* manuscript ID RSPB-2020-1657). We hope this revised manuscript is now clearer to readers who are neither microbiologists nor theoreticians. The attached version indicates the specific changes with **red text**.

Reviewer 3

The question of how social traits evolve in microbes is an important one. Consequently, rigorous evolutionary theory that can be applied to microbial data is a major question. I like the approach of trying out different models on a well-chosen selection of empirical papers. The contribution that the paper makes is the finding that complex models with polynomial and interaction terms and log-transformed fitness yield better fit, suggesting that the fitness consequences of behaviour are more complex than realized. The paradigm of the conflict between canonical and non-linear approaches was not helpful to me. This is more a simple vs. complex model question—see point 1 below. The ‘canonical approach’ is not set in stone—the authors made the decision to use the regression approach designed for a single generation to model longterm selection. They chose the variables to include in the canonical regression and the more complex regression. The paper would be greatly more understandable if they set up their questions more clearly instead of working to convince the reader that they are busting a canon. Essentially I’d like to see this paper rewritten for the non-theorist and non-microbial person, who need much more help in following this paper.

We agree that the biological complexity of fitness effects among interacting microbes is interesting, but that’s not the main point of our manuscript. The main point is that the canonical formulations of social evolution theory are insufficient to meet the analytical needs of empirical microbiologists. We can fit complex fitness models just fine—microbiologists already do that. The problem is that there’s no way to quantitatively connect those results to the general theory of kin and group selection. The point of our manuscript is to explicitly demonstrate that and to identify the ways in which theory and data are poorly matched. The fact that the most general versions of the theory are only designed for a single generation while many microbial interactions span multiple generations (“long term selection” here is typically 24-48 hrs) is one of those disconnects.

We now more explicitly clarify which theory we call canonical and why (line 99).

1. I like how this comes together, but the pieces need to be referenced. Lande and Arnold 1983 introduced an approach for analyzing multiple traits with polynomial and interaction terms, which has been applied to multilevel selections—see multiple papers by Goodnight. Game theory is essentially an analysis of how the fitness consequences of behaviour by one player can depend on the behaviour of the other player.

We now cite multilevel selection analyses that include additional polynomial and interaction terms (line 295).

2. Paragraph line 56. This paragraph is indicating that there is a big problem with evolutionary theory, but only identifies the problem obliquely without actually demonstrating that problem. The last sentence “Or perhaps the weak, additive fitness effects common in canonical theory [3, 2, 25, 9, 26] are more problematic in microbial systems where large populations and rapid reproduction make selection especially effective.” bothers me as the “Or” implies an “Either” that is not obvious. It feels like a rhetorical rather than a scientific argument.

Some participants in the kin/group selection debate do believe there is a big problem with evolutionary theory, but for reasons other than what we’re concerned with here. Our issues are described in the Results and Discussion. In some ways, the whole point of our analysis is to “demonstrate the problem”. In any case, we’ve revised this paragraph to remove the offending “Or” (line 52).

3. Line 60. “We focus on a ‘common mix experiment’ design for assaying the fitness effects of social interactions by measuring how local genotype frequency affects performance” This paper would be considerably more understandable if this sentence were unpacked into a few paragraphs—what is a common mix experiment, why local genotype freq is a measure of social environment, how the phenotypic difference between strains is included in the model. The introduction talks about how fitness is assessed in microbial populations, but not how social behaviour can be measured.

We now more elaborate on mix experiments and what they measure (lines 60, 68). We also explicitly state that we are evaluating kin and group analyses that are formulated entirely in terms of genotypes and fitness with no explicit reference to behavior or other social phenotypes, since these are typically not measured in mix experiments (line 99).

4. Fig 1. I get that this is supposed to demonstrate the problem with theory, but it is not understandable on its own. Microbial mix, Wrightian fitness are not explained, nor are the Kin and multilevel selection models used to create predictions.

We’ve revised the legend to Fig. 1 to be more intelligible on its own. We also note that the first reference to Fig. 1B is in the Results (line 152), by which point mix experiments, Wrightian fitness, and the statistical models used have all been explained.

Theories can’t use models, researchers use models. The lines show the predicted fitness (or fitness ratio) from a model, not the model itself. I have not seen the model when I encounter the figure, so I can’t even begin to know how astonished I should be by predictions.

We now clarify in the legend to Fig. 1 that we are discussing statistical regression models, not the theoretical models that are a concise mathematical descriptions of some biological process. Statistical regression fits an equation to data by adjusting the equations’ parameters. The equations themselves are called statistical models. The versions of kin and group selection theory we address here both use particular statistical models to describe fitness effects. The lines in Fig. 1B show those statistical models fit to this dataset.

1A legend needs to actually say that the white microbes produce a resource (black dots) that both black and white microbes need the resource, and that the fitness is indicated by the number of microbes in the final population (and specify ‘final’). If white microbes are extirpated in the 3rd population, why does it have higher fitness than the 4th population?

The caption to Fig. 1A now clarifies that it is an illustration of how mix experiments work (not an illustration of the biology or results in Fig. 1B). The fitness values for the illustration are shown in Fig. S1.

5. Figure design comments: The contrast between shades of gray for the different lines is hard to identify against ggplot’s default gray background. Total population and non-producer are extremely similar, and from the legend, I expect to find these classes in all 3 figure panels. The points are not jittered so we don’t see the actual data very well.

We now include boxes around the figure legends to clarify that total-group and nonproducer data are plotted in different panels. Initial genotype frequency is a continuous quantitative value, not a categorical one, so it would be inappropriate to jitter points here. The dataset in Fig. 3 only has a few discrete initial frequencies, but many others have a continuous range. Jittering would effectively be plotting inaccurate data and would obscure the functional form of the relationship between fitness and initial frequency.

6. The authors identify the theory they are evaluating in the methods under “applying theory” I would like to see this in the introduction.

Our Introduction sets up the goals of this work (to evaluate kin and group selection theory as tools for quantitative data analysis) and outlines our approach. The specific theoretical formulations we evaluate are more appropriately described in the Methods and Supplementary Material.

7. “Canonical” is not a clearly defined term—see <https://en.wikipedia.org/wiki/Canonical> and beyond these meanings there’s a confusing array of canonical correlation analysis, canonical correspondence analysis, and more in ecological statistics, and so it isn’t immediately clear what the authors mean. I’d recommend using “simple” here.

We now clarify what we mean by “canonical” and why (line 99).

8. As I read this section, the problematic approach (canonical) resulted from how authors themselves initially conceptualized the theory and applied it to microbes, rather than there being any examples of poor practice in the literature. In particular, G is defined as initial strain frequency, a constant, but in this system, genotype frequency will change rapidly.

As we describe in the Introduction, there are very few examples where microbiological authors analyze their data in the same terms as canonical theory. And as we describe in the Supplemental Material, the approach we take here is as direct an application of canonical theory as is possible for these experiments. Canonical theory is typically expressed in discrete time, which would correspond to the initial and final population states. Genotype frequency may change over the course of an experiment, but its initial value does not. G is indeed constant.

9. See Multilevel selection theory and evidence: a critique of Gardner, 2015. C. J. GOODNIGHT. doi:10.1111/jeb.12685 for why not to use a Price equation approach to multilevel selection. The kin selection approach meets Goodnight’s approach to multilevel selection.

There may be good reasons to not use a Price equation approach, but it is a prominent approach that does get used, so we include it in our analysis. And without it we would be unable to make any meaningful comparison between kin and group selection analyses of these experiments. As we already describe on p. 6 of the Supplementary Material, contextual analysis and neighbor-modulated fitness analyses of mix experiments are identical.

10. Line 154. Intermediate optima, trait interactions and selection peaks are also ‘canonical’—e.g. Wright’s fitness landscape.

We now clarify which theory we refer to as “canonical”, and why (line 99). We also describe how including extra terms in the canonical formulation would correspond to intermediate optima, trait interactions, and selection peaks (line 295).

11. Line 197 Mixing frequency? Same as strain frequency?

We now clarify that mixing frequency is the initial strain/genotype frequency in mix experiments (line 84).

12. Fig 2. No metrics are given for these histograms, so we are judging by eye. Give someone unfamiliar with judging models by AIC enough information to evaluate these figures. What would show that ‘linear’ models are better? Can any statistical tests be done? All the results are figures for the reader to evaluate by eye, without much guidance.

We now include a brief orientation to AIC for unfamiliar readers (line 122, Fig. 2). Bar color shows the metric for evaluating AIC. We chose AIC because (1) frequentist statistical tests are only meaningful when comparing nested regression models (where one has a subset of the other’s predictors), (2) we wanted to avoid issues with multiple comparisons, and (3) we’re interested in the distribution of model fits and not just the mean difference.

13. Line 275. If all results were done in `lm()` (calculations and statistics section) they are all linear models and least squares regression. Transformations of variables, polynomials, and interaction are a common part of linear additive models. All the models as described are statistically linear, least squares regressions.

We’ve revised this sentence to say that the main problem with canonical theory (for analysis of microbial data) is that it’s formulated in terms of additive, independent effects on Wrightian fitness (line 288). We suspect that there might be useful formulations that aren’t based on regression analysis, but since that’s not something we discuss here there’s no need to muddy the waters.

14. Line 285. Researchers commonly add a constant to data before log transforming to deal with zeros. Would doing that affect these results?

It's not clear adding a constant would be appropriate for our goals. Most of our results concern absolute effect sizes whose values and units are biological meaningful by themselves (not just a stepping stone to a P -value). What fudge constant would leave that intact? The choice of constant would also have a large effect on the relative goodness-of-fit of linear relative to log-scale models. Presumably one would have to determine which zeroes are biological (*e.g.* zero cells in the final population) and which are merely below the researcher's limit of detection, and treat them differently.

The datasets with zeroes are visible in the Supplemental Figures. In some cases adding a constant would create a region of the data that now gets included in "truncated" form. That'd make models with terms that are nonlinear over log scales an even better fit than we see already.

As we already say in the Discussion, dealing with zeros is a challenge for theory formulated in terms of Malthusian fitness or other log-scale effects. But since the point of this manuscript is to identify the issues with current theory, not to solve them, it seems fine to leave that question open.

Appendix D

Manuscript ID RSPB-2020-1657.R2

Response to reviewers

Please find below our responses to the reviewers' comments on our manuscript, "Evaluating kin and group selection as tools for quantitative analysis of microbial data" (*Proceedings B* manuscript ID RSPB-2020-1657.R2). The attached version indicates the specific changes with **red text**.

Reviewer 3

This manuscript reads much better, and the authors have addressed many of my concerns. However, there is considerable inconsistency in terminology that hindered my ability to follow the arguments. The manuscript needs editing for consistency in terminology for initial genotype frequency, kin selection, and components of multilevel selection. The figures need to be remade to use consistent terminology for initial genotype frequency (G) on the X-axis and to be accessible. The authors must indicate which panels indicate kin selection, the group component of multilevel selection, and the within group component of multilevel selection. Explanations of theory should include the basics and indicate how they are specified for analysis of mix experiments.

I will leave the use of canonical (as opposed to data driven) theory up to the editor. 'Canonical' has connotations related to its use as a synonym for orthodoxy, authority and church law that I find overwhelming, but others may find it more useful. I see the appeal of breaking canon, but I have found that theoreticians are aware of the assumptions and consequent limitations of their theory, and seen that models with weak selection and additive effects generate complex nonlinear predictions.

Responses to my comments

1. Satisfactory
2. Satisfactory – this paragraph reads much better
3. Mostly satisfactory. Fig 1A shows that a typical difference in social behaviour is that one genotype is a producer of an external good that both genotypes use and the other genotype is a non-producer. Being a producer/nonproducer is a behaviour that affects the fitness of neighbours. I suggest including the explanation that these traits that affect the fitness of others in the population because it justifies the use of kin selection/multilevel selection.

One can use still kin and multilevel descriptions of selection even when individuals don't affect each others' fitness, but the "social" fitness terms would be zero.

4. Because many people skim over methods on the first read of a paper, figures should be self-explanatory. Part A is very clear. In part B the legend needs to be more specific. Genotype is ambiguous here, since it could refer to genotype frequency or genotypic value. Suggested wording: B) Estimation of key parameters for kin selection and multilevel selection in a hypothetical dataset with Wrightian fitness and additive effects. The left panel illustrates the analysis of kin selection on the nonproducer strain, with a regression of strain fitness on genotypic value ($g = 0$ for producer genotype and $g = 1$ for non-producer genotype) and mean group genotype (G = initial frequency of nonproducers) (Table S2, Eqn. S6). The middle panel illustrates the group selection component of multilevel selection, with the regression of total group fitness on mean group genotype G (Table S2, Eqn. S9). The righthand panel describes the regression for the estimated value of the within-group difference in fitness (fitness of non-producers minus the fitness of producers) on mean group genotype G (Table S2, Eqn. S9), the initial frequency of non-producers.

In the previous round of revision, reviewer comment 4 concerned Fig. 1. Neither Fig. 1 nor its legend contain the word “genotype”. The reviewer now seems to be referring to Fig. S1. Reasonable people can prefer different amounts of detail in their figure legends—it’s a matter of scientific style. Because this is a supplemental figure illustrating the canonical fitness models described in the text, we’re happy to leave Fig. S1 as is.

5. While the figures are better, authors can do much more with use of colour, shape of points, size of points, and solid vs. dashed lines to make them accessible to all readers.

6. Ok

7. See initial comments above

8. Acceptable. Though initial genotype frequency does not change over the experiment, the terminology the authors use for it does change throughout the paper, and ‘initial’ is often dropped.

9. Satisfactory.

10. Satisfactory

11. The clarification is helpful here, but the best protocol is to settle on consistent terminology and use it throughout.

12. Good. Please add to the orientation the equivalent of “Given a set of candidate models for the data, the preferred model is the one with the minimum AIC value.” (Wikipedia)

We now clarify that the best-performing model is the one with the least AIC (line 126).

13. Good.

14. Ok

Other comments on the manuscript:

15. Line 52-53. Weak selection is often assumed in modelling because strong selection changes genetic variance. It is a simplification that often make modelling tractable. Additive variance is the source of parent offspring correlation.

16. In Figure S2, the “fitness models poorly specified for strong selection and nonadditive effects in microbial data.” are linear as expected. In Fig 1, the “lines show fit of canonical fitness models, whose additive fitness effects” show complex curving. Can the authors provide the models for these examples and explain why they portray ‘additive canonical models’ so differently?

Figures 1 and S2 show the exact same fitness models, but Fig. 1 uses logarithmic scales for the y-axes (as indicated by the tickmark labels). The methods already describe the equations for the canonical fitness models (lines 107, 114).

17. Line 104–110. Refer to table S2 when defining g and G . How the initial Genotype frequency is also G , the mean group genotype, is explained only here.

The beginning of this section already points to Table S2 (line 96). It seems unnecessary to refer to it again six sentences later when we’re still explaining the math. Also, explaining that initial genotype frequency is equal to mean group genotype (for our parameterization of genotypic value) seem appropriate here in the Methods but redundant elsewhere.

18. For the reader’s understanding, label how the equation apply “Malthusian neighbour-modulated fitness (kin selection) with interaction $m \sim g + G + gG$, Malthusian multilevel group selection $M \sim G$, and Malthusian individual selection $\Delta m \sim G$ ”

The reviewer appears to be referring to line 137, which describes the fitness models used to compare fitness effect sizes. It’s not immediately relevant here to reiterate which are neighbor-modulated fitness and which are multilevel fitness. In any case, the relevant results figures already do that (Figs. 3, 4, and others in the supplementary material).

19. Explain this more clearly “slopes are comparable to intercepts because they measure the total effect of group genotype from zero to one.”

In general, slopes are not meaningfully comparable to intercepts because they have different units. Apples to oranges. But in this case the x-axis goes from zero to one, so the slope equals the difference between the line’s value at one and its value at zero, which has the same units as the intercept. Apples to apples.

20. Are mixing frequency (line 170) and local genotype frequency (figure 1 legend) the same as initial genotype frequency? Near eqn S13 and figure legend S4 the term mixing proportion is used. Is this initial gene frequency? Figure axes with initial proportion—is this initial genotype frequency? Near eqn 18 mixing frequency is used. Does proportion differ from frequency in these experiments? If so, use the terms “initial genotype frequency” and/or “mean group genotype” consistently throughout the paper, without introducing more synonyms.

We already describe how “mixing frequency” means the same thing as “initial genotype frequency” (line 84). The introduction also describes how “local genotype frequency” is the thing that mix experiments manipulate (line 61). We use the term “initial proportion” to distinguish this specific measure of relative abundance (n_A/N) from other measures like initial ratio (n_A/n_B), and we already define it in Table S2.

21. The finding of frequency-dependent selection for within-fitness differences is unfortunately confusing, standard as that terminology is. It should be made clear here that when the within-group selection on individual trait genotypic value depends on the group trait “initial genotype frequency” and/or “mean group genotype”, that this means there is frequency-dependent selection.

The manuscript now explicitly states that “frequency-dependent selection within groups” means that within-group relative fitness was a function of initial genotype frequency (line 172).

22. I suggest using ‘kin selection’ generally throughout, with ‘neighbour-modulated fitness approach to kin selection’ used only when this distinction is crucial. Right now they are used synonymously and without any apparent distinction being made. The text around the call to Fig.2 refers to kin selection, but figure 2 itself uses neighbour-modulated fitness.

We like how “neighbour-modulated fitness” reminds readers of what quantities are being described and what they’re a function of: $w(g, G)$. Also, to some readers “kin selection” refers to the whole process of selection and not just the fitness effects. In any case, there doesn’t appear to be any ambiguity. We prefer to leave the text as is.

23. Line 200 spelling of multilevel

Fixed (line 202).

24. The text below eqn. S6, kin selection is explained in sometimes in general terms and sometimes specifically for the mix experiments.

The two sentences following Eqn. S6 describe what the two terms on the right hand side of Eqn. S6 correspond to in a neighbor-modulated fitness analysis of mix experiments.

25. Consider adding an explanatory section of frequency-dependent selection in the supplemental materials.

The manuscript now explicitly states that “frequency-dependent selection” means that within-group relative fitness was a function of initial genotype frequency (line 172).

26. In the supplementary materials section Strain and group fitness outcomes, add the clarification that neighbor-modulated fitness is used to refer to a particular version of kin selection as in line 103.

The text at the beginning of this section now clarifies (again) that neighbor-modulated fitness is one formulation of kin selection.

27. Table S3 needs final figure references.

Fixed.

28. Figure S2 left panel. I can’t distinguish any producer symbols with the dark grey and medium grey on light grey with considerable overlap. With supplementary figures with coloured symbols, the small size of the symbols makes it more difficult to distinguish their colour. There are online tools to help produce figures with accessible contrast.

We’ve lightened the light grey points in Fig. S2 so they’re more distinguishable. The only supplementary figures with symbols of different colors sharing the same plot are the plots showing the analyzed datasets. As far as we are aware, blue and orange are distinguishable to people with both forms of color blindness.

29. Overall the documentation and organization is clear for the data and code. However, `illustrate-issues` does not run for the data example. I tried to see if I could answer point 16 myself. However, the code does not run because a chunk of variable creation code was omitted. The sample data set only contains variables `strain.B` `strain.A`, `initial.number.B`, `initial.number.A`, `final.number.B`, `final.number.A`. The next line asks for a subset with multiple variables not yet created, including `initial.ratio.A`, `initial.proportion.A`, `fitness.A`, `fitness.B`, `fitness.total`, `fitness.ratio.A`. Neither `initial.ratio.A` nor `fitness.ratio.A` were created in the previous hypothetical example. There are also some apparent name changes e.g. `fitness.total.group` vs. `fitness`.

We downloaded the `dryad` package from the reviewer link and were able to successfully run the commands in `illustrate-issues.R` to reproduce Fig. 1B and Fig. S2. Note that the section labelled `ILLUSTRATE ISSUES WITH CANONICAL KIN AND MULTILEVEL REGRESSIONS` reads in a `results/` file, not a `data/` file. The `results/` files were generated by `calculate-fitness.R` and include the variables the reviewer describes as missing.